# Early head-up mobilisation versus standard care for patients with severe acquired brain injury: A systematic review with meta-analysis and Trial Sequential Analysis

Christian Gunge Riberholt[1,2]*, Vibeke Wagner[1], Jane Lindschou[3], Christian Gluud[3], Jesper Mehlsen[4], Kirsten Møller[2,5]

1 Department of Neurorehabilitation, Traumatic Brain Injury, Rigshospitalet, Copenhagen University Hospital, Hvidovre, Denmark, 2 Department of Clinical Medicine, Faculty of Healthcare Sciences, University of Copenhagen, Copenhagen, Denmark, 3 Copenhagen Trial Unit, Centre for Clinical Intervention Research, Rigshospitalet, Copenhagen University Hospital, Copenhagen, Denmark, 4 Surgical Pathophysiology Unit, Juliane Marie Centre, Rigshospitalet, Copenhagen University Hospital, Copenhagen, Denmark, 5 Department of Neuroanaesthesiology, Rigshospitalet, University of Copenhagen, Copenhagen, Denmark

* Christian.riberholt@regionh.dk

**Data Availability Statement:** All relevant data are within the paper and its Supporting Information files.

## Abstract

### Background

There is increasing focus on earlier rehabilitation in patients with traumatic or hypoxic brain injury or stroke. This systematic review evaluates the benefits and harms of early head-up mobilisation versus standard care in patients with severe acquired brain injury.

### Methods

We searched Medline, CENTRAL, EMBASE, four other databases and 13 selected clinical trial registries until April 2020. Eligible randomised clinical trials compared early head-up mobilisation versus standard care in patients with severe acquired brain injury and were analysed conducting random- and fixed-effects meta-analyses and Trial Sequential Analysis (TSA). Certainty of evidence was assessed by GRADE.

### Main results

We identified four randomised clinical trials (total n = 385 patients) with severe acquired brain injury (stroke 86% and traumatic brain injury 13%). Two trials were at low risk and two at high risk of bias. We found no evidence of a difference between early mobilisation vs. standard care on mortality or poor functional outcome at end of the intervention (relative risk (RR) 1.19, 95% CI 0.93 to 1.53; $I^2$ 0%; very low certainty) or at maximal follow-up (RR 1.03, 95% CI 0.89 to 1.21; $I^2$ 0%; very low certainty). We found evidence against an effect on quality of life at maximal follow-up. The proportion of patients with at least one serious adverse event did not differ at end of intervention or at maximal follow-up. For most comparisons, TSA suggested that further trials are needed.

**Funding:** CGR received grants from the The Danish Victims Fund (grant 16–910-00043; http://www.offerfonden.dk/da/GlobalMenu/english.aspx) and The Danish Physical Therapists' Association (15242; https://www.fysio.dk/). The funders had no role in study design, data collection and analysis, decision to publish, or preparation of the manuscript.

**Competing interests:** The authors have declared that no competing interests exist.

## Conclusions

We found no evidence of a difference between early mobilisation versus standard care for patients with severe acquired brain injury. Early mobilisation appeared not to exert a major impact on quality of life. This systematic review highlights the insufficient evidence in patients with severe brain injury, and no firm conclusions can be drawn from these data.

## Trial registration

Protocol uploaded to PROSPERO: April 2018 (revised October 2018, CRD42018088790).

## Background

Severe acquired brain injury is brain damage that occurs after birth and is unrelated to congenital or degenerative conditions [1]. The World Health Organization considers acquired brain injury a major public health problem [2]. It affects people of all ages and infers a large burden on quality of life and health economics [2]. The severity of acquired brain injury is defined in a variety of ways depending on the aetiology. Severe stroke is often defined by a National Institute of Health Stroke Scale score > 16 [3], whereas severe traumatic or anoxic brain injury is characterised by a low Glasgow Coma Score ($\leq$ 8) [4] or for traumatic injury a post-traumatic amnesia period of more than 28 days [5–7].

During recent years, increased focus has been given to early physical intervention within many subspecialties of neurorehabilitation [5,8,9]. Early mobilisation intends to counteract the adverse effects of prolonged bed rest on primarily the cardiovascular and musculoskeletal systems, the internal organs, as well as arousal in patients with chronic disorders of consciousness [10–14]. On the other hand, concerns have been voiced that mobilising the patient head up may reduce cerebral blood flow and/or intracranial pressure, thus negatively impacting functional level [15]. These concerns were not supported by the cluster randomised trial by Anderson et al., who showed no difference in functional outcome after three months when elevating the head of the bed early to 30 degrees compared to participants lying flat in supine positioning [16].

Many clinical guidelines recommend mobilisation of patients with stroke started within the first 48 hours of ictus [17,18]. The effect of early mobilisation versus standard care in patients with stroke was investigated in the AVERT II trial [19], which suggested that early mobilisation lead to earlier return to walking [20]. The subsequent AVERT III trial, however, showed less positive results [5], finding an odds ratio of a favourable outcome for early mobilisation compared with standard care as measured by the modified Rankin scale at three months of 0.73 (95% confidence interval (CI) 0.59 to 0.90) (3). However, some criticism was raised towards this trial. Thus, most of the patients were with mild rather than severe stroke, with around 40% being able to walk independently after disease onset [21]. Also, a secondary analysis of the AVERT III trial suggested another conclusion, i.e. that early but shorter and more frequent mobilisation after stroke seemed to be beneficial compared with standard care when controlling for stroke severity and age [22]. Importantly, such subgroup analyses should only be considered hypothesis-generating and further research is warranted [23].

Guidelines on the management of severe traumatic brain injury do not have recommendations on the use or timing of mobilisation after severe brain injury [24–26]. In a quasi-randomised study on patients with traumatic brain injury, Andelic et al. found less 12-months

disability when comparing an unspecified early rehabilitation regime in the intensive care unit to delayed treatment [8]. However, such non-randomised studies are known to overestimate intervention effects [27]. The beneficial or harmful effects of early mobilisation thus remain incompletely explored in patients with severe acquired brain injury.

## Objectives

This systematic review aimed to assess benefits and harms of early head-up mobilisation, with the head and torso elevated more than 50 degrees above the horizontal level, compared with standard care in patients with severe acquired brain injury.

## Methods

The protocol for this systematic review was submitted to the PROSPERO-database (CRD42018088790) in April 2018 (see Protocol in S1 File) and adheres to the Preferred Reporting Items for Systematic Reviews and Meta-analyses (PRISMA) (Checklist in S1 File) [28].

### Criteria for considering studies for this review

**Types of studies.**  Randomised clinical trials aiming at evaluating benefits and harms of early head-up mobilisation regardless of language, publication date, publication type, or publication status were included. We did not directly search for quasi-randomised studies or observational studies, but such studies were included when encountered during our searches. We reported separately any harms they reported, as they may provide information on rare or late occurring adverse events we could not identify in randomised clinical trials [29,30]. We are aware that the decision not to search systematically for all observational studies may have biased our review towards the assessment of benefits and may overlook certain harms, such as late or rare harms.

**Types of participants.**  We included patients with severe acquired brain injury. Broadly defined, this is an acute injury that is not caused by degenerative processes and was not present at birth. For the present systematic review, acquired brain injury was specifically defined as a direct brain injury caused by one of the following mechanisms:

- major stroke: interruption of blood supply to the brain usually caused by one or more bursting blood vessels (haemorrhagic) or because of blockage of one or more vessels (ischaemic) [31] and associated with a National Institute of Health Stroke Scale (NIHSS) score > 16 [5]; *or*

- severe traumatic brain injury: injury resulting from trauma to the head and any coinciding or subsequent complications, including hypoxia, hypotension, intracranial haemorrhage, and raised intracranial pressure [6] and with a duration of post-traumatic amnesia of more than 28 days or Glasgow Coma Score < 9; or

- severe diffuse hypoxic brain injury: diffuse damage arising from trauma due to a range of other acute incidents including hypoxia (e.g. resulting from drowning, electrocution, anaesthetic accident) [6] and with a duration of post-injury amnesia of more than 28 days or Glasgow Coma Score < 9.

**Types of interventions.**  The characteristic of the intervention of interest was defined as any intervention comparing an early intervention of head and torso mobilisation to at least 50 degrees compared to participants lying flat in supine positioning and comparing this with a

control intervention of mobilisation to less than 50 degrees compared to participants lying flat in supine positioning.

**Types of outcomes.**    All outcomes were assessed at the end of the intervention (as defined by the trials; primary outcome) and at the last follow-up.

*Primary outcomes*

- Mortality or poor functional outcome: This was defined as a poor functional outcome measured on any scale. For the modified Rankin scale (mRS), a poor functional outcome was recorded if the score was from 5 to 6, with 5 being severe disability and 6 being death. For the Disability Rating Scale (DRS), we defined poor outcome as a score from 12 to 30. The DRS has a highest score of 30 (equalling death, with 29 equalling an extreme vegetative state). Finally, for the Functional Independence Measure, we defined a poor outcome as an improvement of less than 0.5 standard deviations derived from the study data.

- Quality of life: This was defined as any variable recording quality of life continuously such as the Australian quality of life (AQoL(4D)) scale, which is a validated measure of quality of life. The score ranges from 1 (best possible quality of life) to 0 (death) to −0.04 (state worse than death) [32]. For this review, we analysed outcome on a continuous scale using mean, standard deviation (SD) and the mean difference between the intervention groups.

- The proportion of participants with serious adverse events, defined as any untoward medical complication that resulted in death; was life-threatening; required hospitalisation or prolongation of existing hospitalisation; resulted in persistent or significant disability; or jeopardised the patient [33].

*Secondary outcomes*

- The proportion of participants with one or more adverse events not considered serious [33].

- The level of consciousness as measured by the Coma Recovery Scale-Revised [34] or other relevant scales as defined in the individual trial.

*Exploratory outcomes*

- Individual serious adverse events.

- Individual adverse events not considered serious.

**Search methods for identification of studies.**    We aimed at identifying all relevant randomised clinical trials, regardless of language or publication status. Selected articles were translated if required.

All reports were uploaded to the Covidence© database for further management [35]. The Covidence© database removed duplicates and managed the selection process, risk of bias assessments, and extraction of data (please see below).

**Database search: Published reports.**    A search strategy for the Medline database was formulated and tested before the first search. The formal search was then performed in Medline (Ovid) (see Search strategy in S1 File) and adjusted to fit the following other databases: Cochrane Central Register of Controlled Trials (CENTRAL, The Cochrane Library); EMBASE (Ovid); CINAHL (EBSCO); PsycINFO; Science Citation Index Expanded on Web of Science; and PEDro. The databases were initially searched in May 2018 and then updated in April 2020. The Boolean search used MeSH terms relating to the condition and the intervention. The intervention term had low specificity in our search as it was our impression that this early intervention is not specifically mentioned in the literature. We used a modified version of the

Cochrane sensitivity-maximising clinical trial filter in the Medline search and adopted it to the other databases except for CENTRAL. We did not use any other limitations in our search.

**Database search: Unpublished or ongoing studies.** We searched for ongoing and unidentified trials on Google Scholar; Database on Research in Stroke (DORIS); The Turning Research into Practice (TRIP) Database; ClinicalTrials.gov; EU Clinical Trial Register; Chinese Clinical Trial Registry (ChiCTR); International Standard Randomised Controlled Trial Number (ISRCTN) registry; Pan African Clinical Trials Registry (PACTR); Australian New Zealand Clinical Trials Registry (ANZCTR); Clinical Trials Registry—India (CTRI); and the World Health Organization (WHO) International Clinical Trials Registry Platform (ICTRP) search portal.

The references of included trials were screened to identify further trials of interest.

## Data collection and analysis

### Selection of studies

All titles and abstracts were screened by VW and either CGR or JM, using the above-mentioned inclusion criteria. Any disagreement between authors was solved by discussion; if any uncertainty remained, the study was included for full-text assessment. Full-text papers were obtained and read by CGR and VW independently; clinical trials to be included were identified based on study type, types of participants, and intervention. The studies were then classified as either eligible, not eligible, or uncertain. Studies that both authors had classified as not eligible were excluded and studies classified as eligible were included. Studies classified as uncertain were discussed between CGR and VW, and additional information was retrieved from corresponding authors of the trials. If individual patient data were not already made available, the corresponding authors were asked to supply data for data extraction for those patients with severe brain injury as defined in our inclusion criteria. Multiple publications on the same trial were analysed as one trial.

### Data extraction and management

All data extraction was done independently by CGR and VW using a standardised data-extraction checklist set-up in Covidence©. CGR is the first author of one included trial [36]. Therefore, data extraction of this trial was assessed by VW and JL.

We extracted the following data:

- General information: publication status, title, authors' names, source, country, contact address, language of publication, year of publication, duplicate publication; trial characteristics: design and setting.

- Interventions: type of intervention used for mobilisation, dose, duration, type of control intervention; participants: inclusion and exclusion criteria, number of participants randomised in intervention and control groups, participant demographics such as sex and age, and baseline characteristics for patients relevant for subgroup analysis.

- Outcomes: number of patients analysed for each outcome. For details, please see the primary and secondary outcome measures section above.

- Risk of bias: please see the risk of bias (quality) assessment below.

- Data relevant for subgroup and sensitivity analyses; for details, please see "Subgroup analysis and investigation of heterogeneity" below.

After data extraction, the Covidence© extraction form was compared by the two authors to ensure detailed and correct extraction. Subsequently, all information was transferred from Covidence© to Review Manager [37].

## Assessment of risk of bias in included studies

Two authors (CGR and VW) assessed all included studies using the Risk of Bias tool ver. 1.0 from The Cochrane Collaboration [38]. CGR is the first author of one included trial [36]. Therefore, the risk of bias of this trial was assessed by VW and JL. We evaluated the following study characteristics: random sequence generation, allocation concealment, blinding of participants and treatment providers, blinding of outcome assessors, incomplete outcome data, selective outcome reporting, and other bias. Furthermore, the domains 'blinding of outcome assessment', 'incomplete outcome data', and 'selective outcome reporting' were assessed for each outcome result. Finally, the overall risk of bias assessment was dichotomised into high (high or unclear) or low.

The observational studies did not undergo formal risk of bias assessment. We are aware that the observational data carry a high risk of selection bias. However, if observational studies do report harms, then in all likelihood harms occurring is at least at that level or above [29,30].

## Measures of treatment effect

Treatment effects were analysed using the statistical programs Review Manager 5 [37], SAS/STAT software [39], and Trial Sequential Analysis (TSA) [40]. For the primary outcomes and adverse events considered not serious, data were dichotomised and analysed as the relative risk (RR) with 95% CI. Individual events reported was presented with a RR and 95% CI. If no event existed for one of the groups, we employed continuity correction with 0.5 in the Trial Sequential Analysis program [40]. When analysing data from single trials, we checked the P values with Chi-square if outcome data from both intervention groups were reported or with Fischer's exact test if one group had zero outcomes. All serious adverse events were reported. As different outcomes measures were used in different trials, we chose to dichotomise these scales and to classify outcomes as either 'good' or 'poor' (please refer to Types of outcome measures section). Level of consciousness was analysed as a continuous outcome using mean difference and minimal clinically relevant differences of 0.5 standard deviations calculated from the observed variance of the trials.

## Sensitivity analysis

The meta-analysis was conducted using random-effects and fixed-effect models as sensitivity analyses); the most conservative result was reported, using a *P*-value for the primary outcomes of 2.5% as significant [41]. For the primary outcome 'mortality or poor functional outcome' we did a sensitivity analysis for mortality and for poor functional outcome among survivors at both time points.

## Trial sequential analysis

Cumulative meta-analyses are at risk of producing random errors due to sparse data and/or multiple testing of accumulating data [38,42–47]. TSA can be applied to control for random errors and assess the risks of imprecision (http://www.ctu.dk/tsa/) [40,41,48]. Similar to a sample size calculation in a randomised clinical trial, TSA calculates the required information size or meta-analytic sample size (i.e. the number of participants needed in a meta-analysis to detect or reject a certain intervention effect reliably) to control random errors [47,49]. The

required information size for a dichotomous outcome takes into account the event proportion in the control group, the assumption of a plausible risk ratio (RR) reduction, and the heterogeneity of the meta-analysis [49,50]. TSA with Lan-DeMets' stopping boundaries enables testing for significance to be conducted each time a new trial is included in the meta-analysis. Based on the required information size, trial sequential monitoring boundaries for benefit, harm, and futility can be constructed. This enables one to determine the statistical inference concerning cumulative meta-analysis that has not yet reached the required information size [46,47]. Firm evidence for benefit or harm may be established if the trial sequential monitoring boundary is crossed before reaching the required information size, in which case further trials may turn out to be superfluous. In contrast, if the boundary is not surpassed, one may conclude that it is necessary to continue with further trials before a certain intervention effect can be detected or rejected. Firm evidence for lack of the postulated intervention effect can also be assessed with TSA. This occurs when the cumulative Z score crosses the trial sequential monitoring boundaries for futility.

For dichotomous outcomes, we estimated the required information size based on the proportion of patients with an outcome in the control group, a relative risk reduction (RRR) of 20%, an alpha of 2.5% for primary outcomes and 3.33% for secondary outcomes, a beta of 10%, and the variance suggested by the trials in a random-effects meta-analysis (diversity-adjusted required information size (DARIS)) [41,49,51]. Additionally, we calculated diversity-adjusted TSA CI. In case there was some evidence of the effect of the intervention, a supplementary TSA was planned based on an anticipated intervention effect equal to the limit of the CI closest to 1.00 [41].

For continuous outcomes, we were unable to identify valid previous data on effect sizes on quality of life, so we chose SD/2 as the anticipated intervention effect [52]. Hence, we estimated the required information size based on the SD observed in the control group of trials at low risk of bias or lower risk of bias and a minimal relevant difference of the observed SD/2, an alpha value of 2.5% for primary outcomes and 3.33% for secondary outcomes, a beta of 10%, and the variance suggested by the trials in a random-effects meta-analysis (DARIS) [41,49]. Additionally, we calculated the diversity-adjusted TSA CI. In case there was evidence of the effect of the intervention, a supplementary TSA was planned to be used based on an anticipated intervention effect equal to the limit of the CI closest to 0.00 [41].

## Assessment of heterogeneity

The statistical heterogeneity was examined between trials using the $I^2$ statistic. Considerable heterogeneity was defined as an $I^2$ between 75% and 100%, substantial heterogeneity between 50% and 90%, moderate heterogeneity between 30% and 60%, and no or low heterogeneity (might not be important) between 0% and 40% [38].

## Subgroup analysis and investigation of heterogeneity

We categorised the results of included studies according to the following considerations:
   Methodological

- Trials at low risk of bias compared to trials at high risk of bias.

   Clinical

- Type of injury (stroke patients, traumatic brain injury, or diffuse acquired hypoxic brain injury).

- Type of mobilisation intervention used (tilt-table intervention compared to other experimental interventions).

- Duration of the intervention period (studies with long duration were defined as those with a duration above the median time and were compared to those with a duration below the median time).

- Intensity of the intervention (studies with high intensity were defined as those with an exercise duration of more than one hour per day and were compared to those with a duration of one hour or less per day).

- Frequency of the intervention (studies with a high intensity frequency were defined as those with four or more intervention sessions per day during the intervention period and were compared to those with three or less sessions per day).

- Timing of the intervention (studies in which the intervention was started earlier than 48 hours after brain injury, compared to those in which it was started later than 48 hours after the brain injury).

### GRADE

A summary of findings table was produced summarising the results of the trials at overall low risk of bias and for all trials, separately. The quality of the available evidence was downgraded if the risk of bias evaluation found evidence of publication bias, heterogeneity, imprecision, or indirectness (e.g. surrogate outcomes) [53,54]. We compared the imprecision assessed according to GRADE using our plausible parameters with that of TSA (i.e. the diversity-adjusted TSA CI) [55,56]. Imprecision was downgraded in GRADE (please see below) by two levels if the accrued number of participants was below 50% of the DARIS, one level if between 50% and 100% of DARIS, and no downgrade if the cumulative Z-curve reaches benefit, harm, futility, or DARIS. Each outcome was evaluated as critical, important or not important as recommended by the GRADE guidelines [57].

## Results

### Description of studies

**Results of the search.** After the removal of duplicates, the initial literature search revealed 13,480 records (for Preferred Reporting Items for Systematic Reviews and Meta-Analyses (PRISMA) flow diagram, please refer to Fig 1). 13,393 records were excluded initially because their title or abstract indicated the study was not related to acquired brain injury or the intervention was not within the scope of the present study. Accordingly, a total of 87 full-text articles were retrieved for full-text assessment. Of these, 82 were excluded (S1 Table in S1 File). One trial was still ongoing, but it was possible to include data from the trial as the data analysis was finished during the review process [36].

Of the 87 full-text articles, 17 primary investigators were contacted to verify if their trial suited our inclusion criteria or for further extraction of patient data. Eleven trialists responded, resulting in the conclusion that six of the trials did not fit our inclusion criteria [58–63]. Six trialists and their affiliated institutions did not respond [64–69]. Therefore, these trials are awaiting classification. We have not been able to retrieve further information from these studies to clarify the eligibility of them and they were not included in the analysis. This left four randomised trials for inclusion in the review [5,19,36,70] and one observational study [71]. Data from the latter study was retrieved from the primary investigator [66].

**Included studies.** We included four trials with a total of 385 participants [5,19,36,70] (Table 1). The included patients had a severe traumatic brain injury (n = 50), severe anoxic

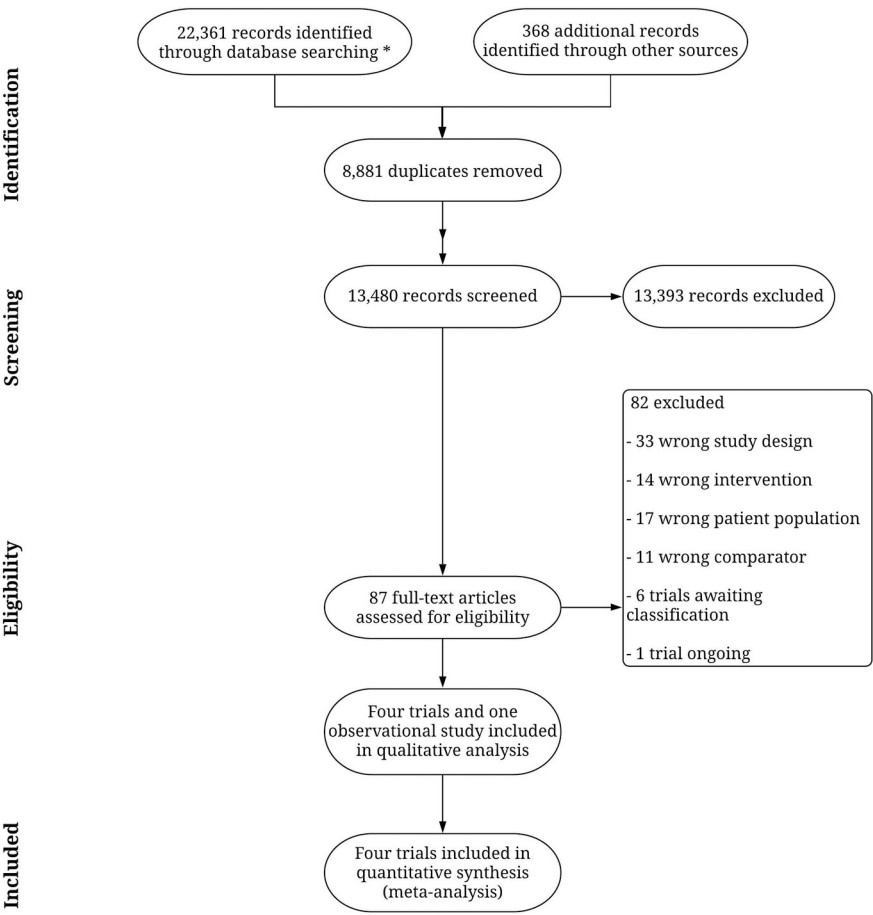

**Fig 1. Flow chart.** Study flow chart through the systematic review. *Detailed search history can be found in S1 Table in S1 File.

brain injury (n = 3), severe other brain injury (n = 1), or severe stroke (n = 331). In total, patients with stroke represented 86% of the population in this review. One trial from the latter category was much larger than the remaining trials (n = 291) [5]. One trial only included patients with severe traumatic brain injury [36]. These two trials had a maximal follow-up of three months [5] and one-year [19]. The trial including patients with traumatic brain injury had a maximal follow-up of one year [36] and the last trial had a maximal follow-up of approximately 4 to 5 months [70].

The intervention was categorised as either mobilisation to the edge of the bed and standing/walking [5,19], or tilt-table mobilisation [36,70]. Two trials performed early intensive mobilisation with a high daily frequency of out of bed mobility within 24 hours versus standard care [5,19]. The other two trials mobilised patients in the intervention group on a tilt table daily starting as early as possible and performed standard mobilisation in the control group [36,70]. In the latter, the experimental intervention was applied at a later stage (mean 14 ± 6 days from injury when studies are combined).

One observational cohort study was included for reporting of harms [71].

**Excluded studies.** We excluded 84 studies as described in the Characteristics of excluded studies (S1 Table in S1 File). The reasons for exclusion were that the study was not a randomised clinical trial or observational study, that the intervention did not include mobilisation

**Table 1. Characteristics of included randomised clinical trials.**

| Trial | AVERT II | AVERT III | Frazzitta et al. | Riberholt et al. |
|---|---|---|---|---|
| Year | 2008 | 2015 | 2016 | 2018 |
| Trial characteristics | RCT | RCT | RCT | RCT |
| Number of trial sites | 2 | 56 | 1 | 1 |
| Intervention used | Out of bed mobilisation | Out of bed mobilisation | Erigo tilt table | Erigo tilt table |
| Criteria for inclusion | > 18 years. First or recurrent stroke Admitted within 24 hours of symptom. Able to react to verbal commands (but did not need to be fully alert). Systolic blood pressure between 120 and 220 mmHg, Oxygen saturation >92%. Heart rate between 40 and 100 beats per minute. Temperature 38.5˚C. | > 18 years. First or recurrent stroke. Admitted within 24 hours of symptom. Treatment with rtPA was allowed. | > 18 years. GCS ≤ 8 for ≥ 24h from the event. VS or MCS on third day after injury. Arterial O2 pressure/O2 flux ratio ≥ 250. Stable hemodynamic | > 18 years. GCS < 11 at inclusion. Tentative diagnose of prolonged VS or MCS. No fractures in lower extremities. Intracranial pressure < 20 mmHg for 24 hours |
| Population | Stroke (n = 4 haemorrhagic) | Stroke (n = 35 haemorrhagic) | Stroke (n = 22, 20 haemorrhagic), traumatic brain injury (14), anoxic brain injury (3) | Traumatic brain injury |
| **Participants** | | | | |
| *Early mobilisation* | | | | |
| Number of participants | 10 | 147 | 20 | 19 |
| Age | 78 (SD ±11) | 77.1 (IQR: 67.7;82.3) | 53 (SD ±15) | 47.8 (18.1) |
| Sex (male) | 5 | 83 | 9 | 13 |
| First stroke or head injury | 5 | 129 | Not reported | 19 |
| Severity | NIHSS 22 (IQR: 19 to 23) | NIHSS 20 (IQR: 18 to 23) | GCS: 7.0 (IQR: 4 to 8) | GCS: 6 (IQR: 4 to 7) |
| *Standard care* | | | | |
| Number of participants | 7 | 144 | 20 | 19 |
| Age | 76 (SD±6) | 74.6 (IQR: 66.6;82.1) | 69 (SD±16) | 41.8 (SD ±18.3) |
| Sex (male) | 3 | 91 | 11 | 14 |
| First stroke or head injury | 5 | 115 | Not reported | 19 |
| Severity | NIHSS 21 (IQR: 18 to 22) | NIHSS 21 (IQR: 18;24) | GCS: 8.5 (IQR: 6.3,10.0) | GCS: 6 (IQR: 4;9) |
| **Interventions** | | | | |
| Degree of elevation | To sitting or standing (90 degrees) | To sitting or standing (90 degrees) | 60 degrees | 70 degrees |
| Dose of mobilisation in early mobilisation group | 40.9 minutes (±31.2) | 186 minutes (IQR: 65;375) | 450 minutes (15 sessions of 30 minutes) | 10.7 ±5.9 times of 20 minutes sessions |
| Dose of mobilisation in standard care group | 12.3 minutes (±9.2) | 102 minutes (IQR: 32;162) | 0 minutes | 0 minutes |
| Time to first mobilisation in early mobilisation group | 21.5 hours (IQR: 16 to 27) | 20 hours (IQR: 13;23) | 12.4 days (SD ±7.3) | 15 days (IQR: 11;16) |
| Time to first mobilisation in standard care group | 35 hours (IQR: 20 to 95) | 29 hours (IQR: 22;43) | 25.1 days (SD ±11.2) | 12 days (IQR: 10;18) |
| **Outcomes** | | | | |
| Death | mRS | mRS | Incident reported | Incident reported as serious adverse event |

(*Continued*)

**Table 1.** (Continued)

| Trial | AVERT II | AVERT III | Frazzitta et al. | Riberholt et al. |
|---|---|---|---|---|
| **Functional outcome** | mRS | mRS | DRS | FIM |
| **Quality of life** | AQoL (4D) | AQoL (4D) | Not measured | Not measured |
| **Adverse events** | Registered SAE and AE (predefined) for three months | Registered SAE and AE during intervention and IME (predefined) for three months | Registered SAE and AE during intervention | Retrospective analysis of all serious and non-serious adverse events during intervention period |
| **Level of consciousness** | Not measured | Not measured | CRS-R | CRS-R |

rtPA: recombinant tissue plasminogen activator; GCS: Glasgow Coma Scale; VS: vegetative state; MCS: Minimally conscious state; IQR: inter quartile range; SD: Standard deviation; NIHSS: National Institute of Health Stroke Scale; mRS: modified Rankin Scale; DRS: Disability Rating Scale; FIM: Functional Independence Measure; AQoL(4D): Assessment of quality of life (4D); SAE: Serious adverse event; AE: Adverse event not considered serious; CRS-R: Coma Recovery Scale-Revised

head up to at least 50 degrees, that the comparator included head up mobilisation to at least 50 degrees (or this could not be ruled out), or that the patient population did not comprise patients with severe acute brain injury. Also, studies, with a broader defined patient population, where less than 10 of the included participants matched our inclusion criteria, were excluded.

**Risk of bias in included studies.** All four included trials were at risk of bias (Figs 2 and 3). Given the exercise nature of the intervention, it was not possible to blind participants nor the persons that delivered the intervention. Some effort at single blinding was done in the AVERT trials; thus, the patients were only informed that they were given one of two rehabilitation approaches without explaining the details of the intervention. Furthermore, all interventions were carried out behind a curtain to avoid unblinding of the remaining investigators, staff, or family [5,19]. The trial by Frazzitta et al. [70] was registered as having an unknown risk of bias with regard to selective outcome reporting, since the trial was registered in clinicaltrials.org only after patient inclusion was completed. The trial used fixed block sizes of 4 and did not blind patients and personnel to the allocation, which increases the risk of selection bias and performance bias. Finally, differences in age and the occurrence of hypertension between the two groups could directly influence the results of the trial. The trial by Riberholt et al. [36] was at risk of bias regarding lack of blinding for the intervention both for the included patients, the staff, and for outcome assessors for some of the outcomes (functional scales). Furthermore, the study was at risk of attrition bias due to incomplete outcome data at follow-up.

## Effects of interventions

**Primary outcomes.** *Mortality or poor functional outcome*. Three trials reported on mortality and poor functional outcome at the end of intervention [19,36,70] and all four trials reported on mortality and poor functional outcome at maximal follow-up [5,19,36,70]. At the end of the intervention, 36 (80%) patients died or had a poor functional outcome in the early mobilisation group versus 31 (67%) in the standard care group. The fixed-effect meta-analysis showed no difference between groups (RR 1.19, 95% CI 0.93 to 1.53; $I^2$ 0%) (Fig 4A) and the TSA-adjusted CI was 0.43 to 3.29 [GRADE certainty VERY LOW] (Table 2). The TSA showed that only 14% of the required information was accrued (S1 Fig in S1 File). At maximal follow-up, 121 (63%) patients died or had a poor functional outcome in the early mobilisation group compared with 114 (60%) for the standard care group. Fixed-effect meta-analysis showed no difference between the two treatment groups (RR 1.03, 95% CI 0.89 to 1.21; $I^2$ 0%) (Fig 5A) and the TSA-adjusted CI was 0.78 to 1.38 [GRADE certainty VERY LOW] (Table 2). The

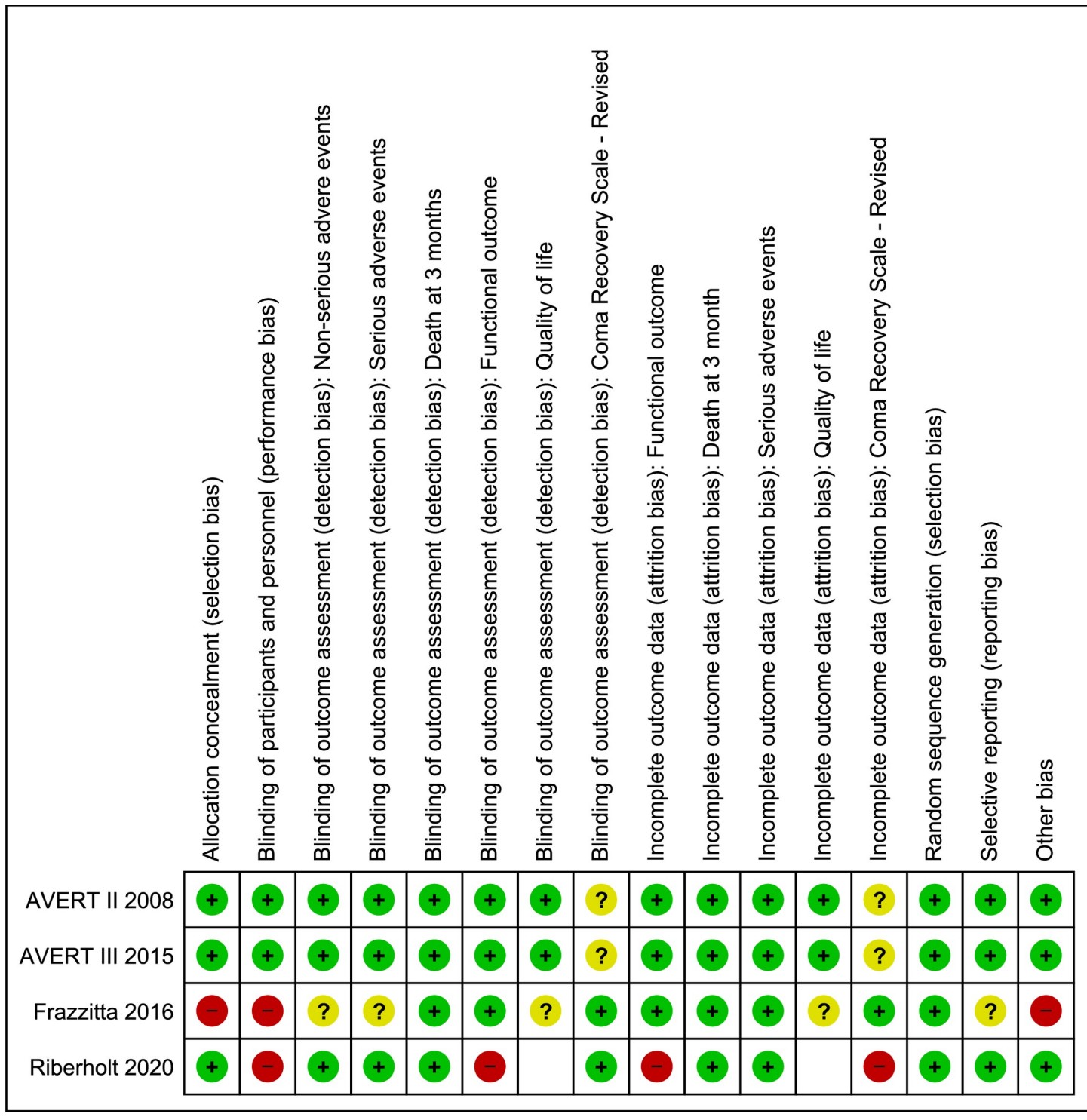

**Fig 2. Risk of bias assessment: Review authors' judgements about each risk of bias item for each included study.**

accrued information size from the TSA at maximal follow-up was too small to reject a 20% RRR achieved by early mobilisation (S2 Fig in S1 File).

Subgroup analysis in patients with stroke alone compared to patients with non-stroke acquired brain injury showed no difference between groups in mortality or poor functional

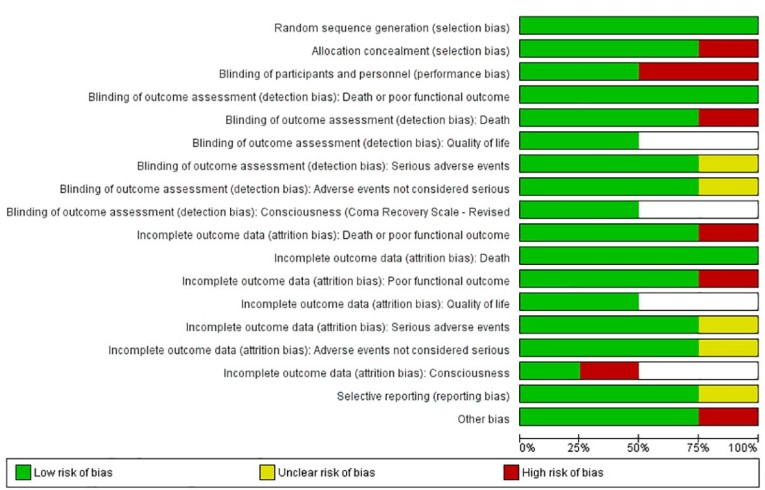

**Fig 3. Risk of bias graph: Review author's judgements about each risk of bias item presented as percentages across all included studies.**

outcome either at the end of the intervention or at maximal follow-up with moderate to low heterogeneity (Figs 4B and 5B).

We found no evidence for a difference between the early mobilisation group versus standard care on mortality at the end of the intervention (RR 1.32, 95% CI 0.88 to 1.98; $I^2$ 0%) (Fig 6A) with the TSA-adjusted CI from 0.25 to 6.89 (Table 2) and severe lack of required information (S3 Fig in S1 File). At maximal follow-up there was no between group difference in mortality (RR 1.26, 95% CI 0.93 to 1.72; $I^2$ 0%) (Fig 7A) and the TSA-adjusted CI between 0.36 to 4.47 (Table 2). The TSA showed a severe lack of required information at maximal follow-up as more than 3,000 patients were needed (S4 Fig in S1 File). Subgroup analysis on patients with stroke alone or patients with non-stroke acquired brain injury also showed no difference at the end of the intervention or maximal follow-up (Figs 6B and 7B).

We found no evidence of a difference between the early mobilisation group versus standard care on poor functional outcome (among survivors) at the end of the intervention (RR 1.22, 95% CI 0.89 to 1.68; $I^2$ 0%) (Fig 8A) and the TSA-adjusted CI was 0.32 to 4.41 (Table 2). The TSA estimated an inclusion of 811 patients in the analysis to reach the required information size (S5 Fig in S1 File). Poor functional outcome (among survivors) at maximal follow up had a RR of 1.12 (95% CI 0.84 to 1.49; $I^2$ 0%) (Fig 9A) and the TSA-adjusted CI was 0.35 to 3.53 (Table 2). Poor functional outcome (among survivors) did not reach the boundaries of benefit, harm, or futility in the TSA with estimated required information size of 1867 patients at maximal follow-up (S6 Fig in S1 File).

**Quality of life.** Two trials reported quality of life at maximal follow-up [5,19]. We found no evidence of a difference between the early mobilisation group versus standard care with a mean difference of 0.0 points (95% CI -0.05 to 0.05; $I^2$ 0%) and the TSA-adjusted CI was -0.2 to 0.2 [GRADE certainty MODERATE] (Fig 10A and S7 Fig in S1 File). The TSA reached futility and beyond the required information size indicating that a minimum relevant difference of 0.1 points is not likely to be found.

**Serious adverse events.** All four trials reported serious adverse events at the end of the intervention. The AVERT II trial reported events for three months by categorically searching for prespecified adverse events [19]. The AVERT III trial reported adverse events during the intervention period (14 days) and reported important medical events for the ensuing three

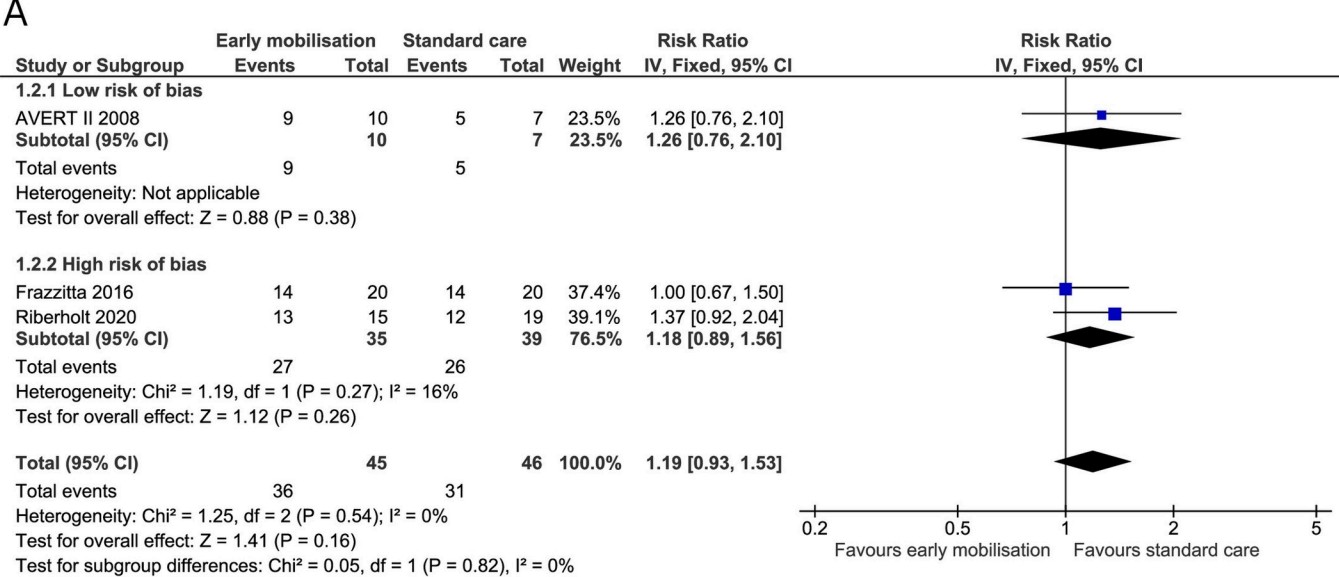

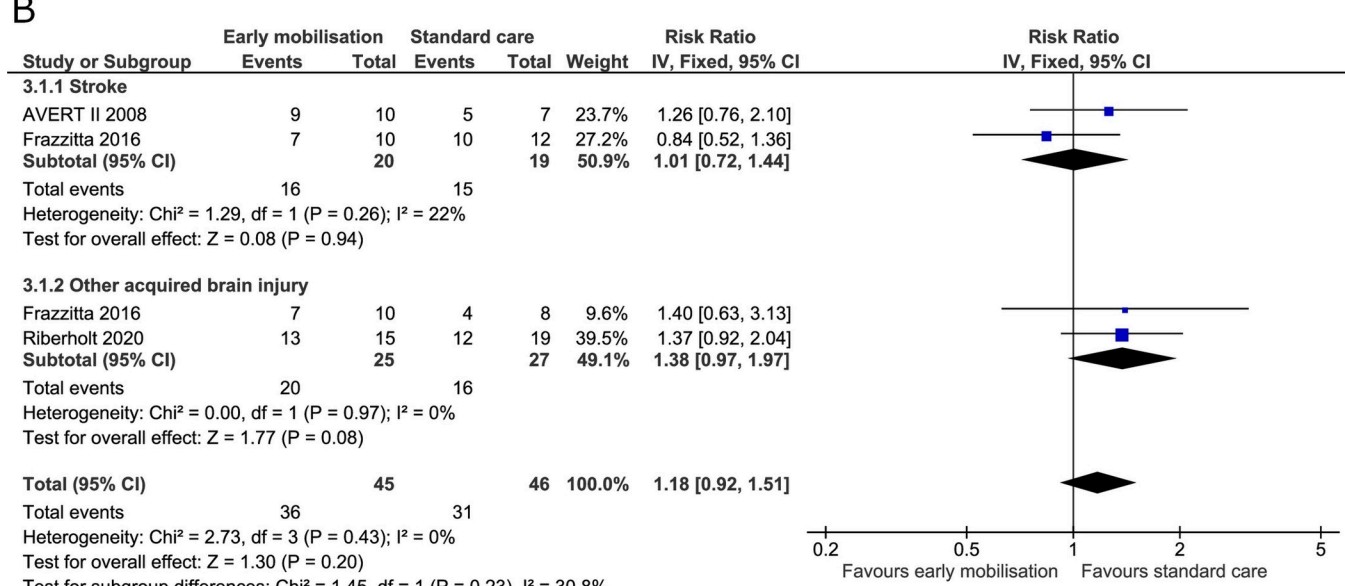

**Fig 4. Comparison of early mobilisation versus standard care–mortality or poor functional outcome at the end of the intervention.** Fig 4A and 4B. Forest-plots showing the results from the fixed-effect meta-analysis of the primary composite outcome mortality or poor functional outcome at the end of intervention with subgroup divided according to risk of bias (A) or diagnosis (B).

months (also prespecified) [5]. Frazzitta et al. reported that they experienced no adverse event during the three-week intervention period in the critical care unit. The trial did, however, report deaths during the intervention period, and we included these as serious adverse events in the analysis [70]. In the trial by Riberholt et al., patient reports were retrospectively screened for serious adverse events by two blinded investigators during the intervention period (up to four weeks) [36]. We found no evidence of a difference between early mobilisation and standard care on serious adverse events at the end of the intervention (RR 1.10, 95% CI 0.86 to 1.39) (Fig 11A) with the TSA-adjusted CI from 0.41 to 3.12 [GRADE certainty VERY LOW].

**Table 2. Results of trial sequential analysis of early mobilisation versus standard care.**

| Outcome | No. of trials | Pc | RRR | MIREDIF / variance | D² | DARIS* | % of DARIS obtained | TSA boundaries crossed? | | TSA-adjusted CI |
|---|---|---|---|---|---|---|---|---|---|---|
| | | | | | | | | Superiority boundaries | Futility boundaries | |
| **Primary outcomes** | | | | | | | | | | |
| Mortality or poor functional outcome at the end of intervention | 3 | 67.4% | 20% | NA | 0% | 652 | 14% | No | No | 0.43 to 3.29 |
| Mortality | 4 | 17.4% | 20% | NA | 0% | 5415 | 7% | No | No | 0.25 to 6.89 |
| Poor functional outcome | 4 | 61.5% | 20% | NA | 0% | 811 | 9% | No | No | 0.32 to 4.41 |
| Mortality or poor functional outcome at the longest follow-up | 4 | 60.3% | 20% | NA | 0% | 848 | 45% | No | No | 0.78 to 1.38 |
| Mortality | 4 | 26.3% | 20% | NA | 0% | 3242 | 12% | No | No | 0.36 to 4.47 |
| Poor functional outcome | 4 | 40.3% | 20% | NA | 0% | 1867 | 14% | No | No | 0.35 to 3.53 |
| QOL at the longest follow-up | 2 | NA | NA | 0.1 / 0.04 | 0% | 199 | 105% | No | Yes | -0.2 to 0.2 |
| Proportion of participants with at least one SAE at the end of intervention | 4 | 35.8% | 20% | NA | 0% | 2115 | 18% | No | No | 0.41 to 3.12 |
| Proportion of participants with at least one SAE at the longest follow-up | 2 | 48.3% | 20% | NA | 0% | 1308 | 24% | No | No | 0.53 to 2.42 |
| **Secondary outcomes** | | | | | | | | | | |
| Proportion of participants with at least one AE considered not serious at the end of intervention | 4 | 41.6% | 20% | NA | 0% | 1574 | 25% | No | No | 0.61 to 1.67 |
| Proportion of participants with at least one AE considered not serious at the longest follow-up | 2 | 39.7% | 20% | NA | 0% | 3278 | 9% | No | No | 0.39 to 3.20 |
| Level of consciousness CRS-R at the end of intervention | 2 | NA | NA | 3.9 / 60.2 | 80% | 766 | 8% | No | No | -33.6 to 33.6 |
| Level of consciousness CRS-R at the longest follow-up | 2 | NA | NA | 3.5 / 50.2 | 70% | 545 | 10% | No | No | -21.57 to 22.81 |

No: number; RRR; assumed relative risk reduction (dichotomous outcomes); Pc: proportion in control group; MIREDIF: minimal relevant difference; SD: standard deviation; D²: diversity (squared); DARIS: diversity-adjusted required information size; QOL: quality of life; SAE: serious adverse events; AE: adverse event.

*α-level (type 1 error) of 2.5% and β-level (type 2 error) of 10% used in calculation of DARIS.

The TSA showed that 18% of the required information size was obtained and the boundaries of benefit, harm, or futility were not surpassed (S8 Fig in S1 File). The estimate did not change after subdividing the patients according to diagnosis (Fig 11B). For serious adverse events at maximal follow-up, the RR was 1.08 (95% CI 0.87 to 1.35) (Fig 12A) and the TSA-adjusted CI was 0.53 to 2.42 [GRADE certainty VERY LOW] (S9 Fig in S1 File). The TSA showed that only 24% of the required information size was obtained and the boundaries of benefit, harm, or futility were not surpassed.

## Secondary outcomes

**Consciousness.** Two trials measured the level of consciousness using the Coma Recovery Scale-Revised at the end of the intervention and maximal follow-up [36,70]. The I² was 80% and 65% at the end of the intervention and maximal follow-up, respectively, indicating moderate to substantial heterogeneity. We found no evidence of a difference between groups at the end of the intervention (mean difference -0.00 points, 95% CI -8.23 to 8.23; I² 80%) (Fig 13A) and the TSA-adjusted CI from -33.60 to 33.60 [GRADE certainty VERY LOW] (Table 2). The cumulative Z score did not reach the boundaries of benefit, harm or futility and would require an information size of 816 patients (S10 Fig in S1 File). The subgroup analysis between patients

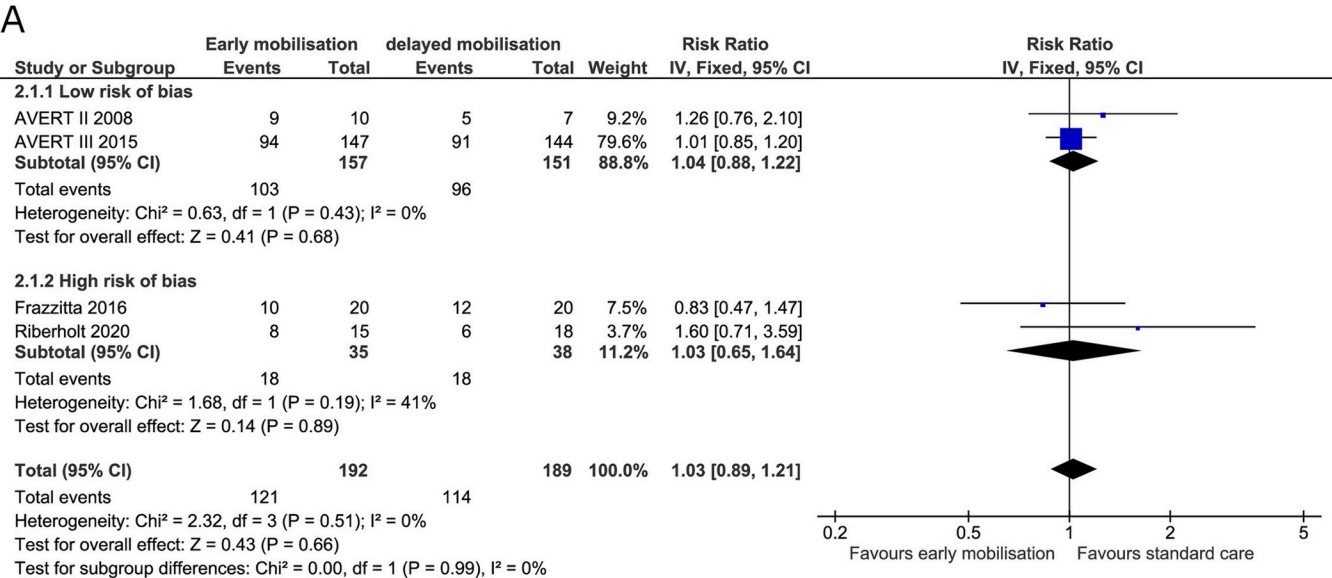

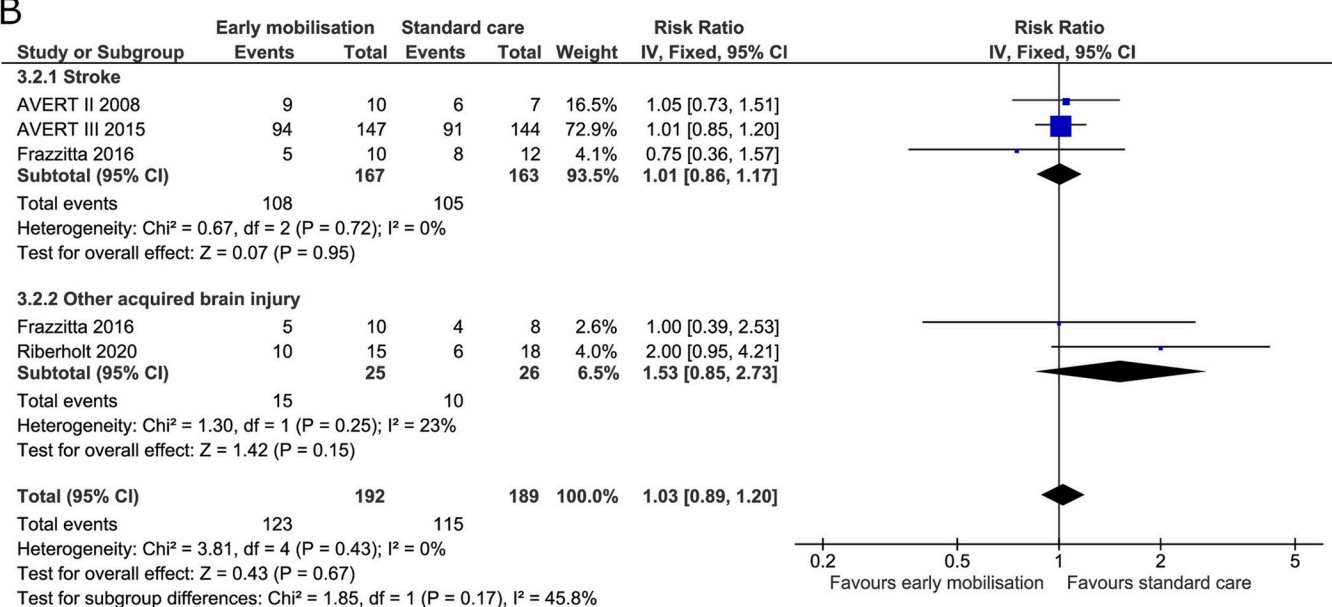

**Fig 5. Comparison of early mobilisation versus standard care–mortality or poor functional outcome at maximal follow-up.** Fig 5A and 5B. Forest-plots showing the results from the fixed-effect meta-analysis of the primary composite outcome mortality or poor functional outcome at maximal follow-up with subgroup divided according to risk of bias (A) or diagnosis (B).

with stroke or other acquired brain injury showed no difference between groups (mean difference 1.38, 95% CI -6.57 to 9.33; $I^2$ 88%) and considerable heterogeneity (Fig 13B). At maximal follow-up, a mean difference of 0.62 (95% CI -4.82 to 6.06; $I^2$ 65%) was found (Fig 14A) and the TSA-adjusted CI from -21.57 to 22.81 [GRADE certainty VERY LOW]. Only 10% of the TSA required information size was accrued (Table 2).

**The proportion of participants with one or more adverse events not considered serious.** Non-serious adverse events were reported in four trials at the end of the intervention [5,19,36,70]. The fixed-effect meta-analysis revealed no difference between early mobilisation and standard care at the end of the intervention (RR 1.02, 95% CI 0.88 to 1.19; $I^2$ 0%) and no

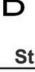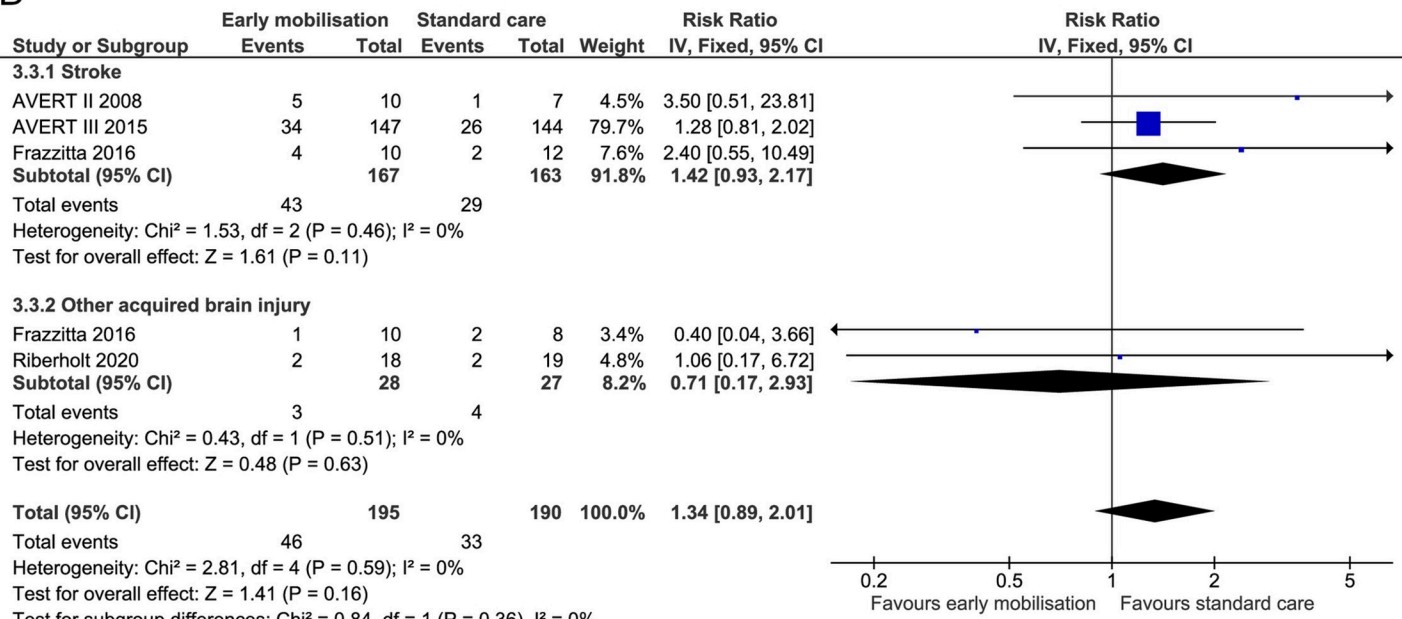

**Fig 6. Comparison of early mobilisation versus standard care–mortality at the end of the intervention.** Fig 6A and 6B. Forest-plots showing the results from the fixed-effect meta-analysis of the outcome mortality at end of intervention with subgroup divided according to risk of bias (A) or diagnosis (B).

heterogeneity (Fig 15A and 15B). The TSA-adjusted CI ranged from 0.68 to 1.50 (Table 2) with the cumulative Z score reaching 23% of the required information size (S12 Fig in S1 File). At the maximal follow-up, only two trials reported data [5,19]. There was no difference between groups at this time (RR 1.07, 95% CI 0.84 to 1.37; $I^2$ 30%) but moderate to low heterogeneity was found (Fig 16A). Furthermore, the TSA-adjusted CI for adverse events not considered serious was between 0.39 and 3.20 at maximal follow-up; only 9% of the required information size was obtained (Table 2).

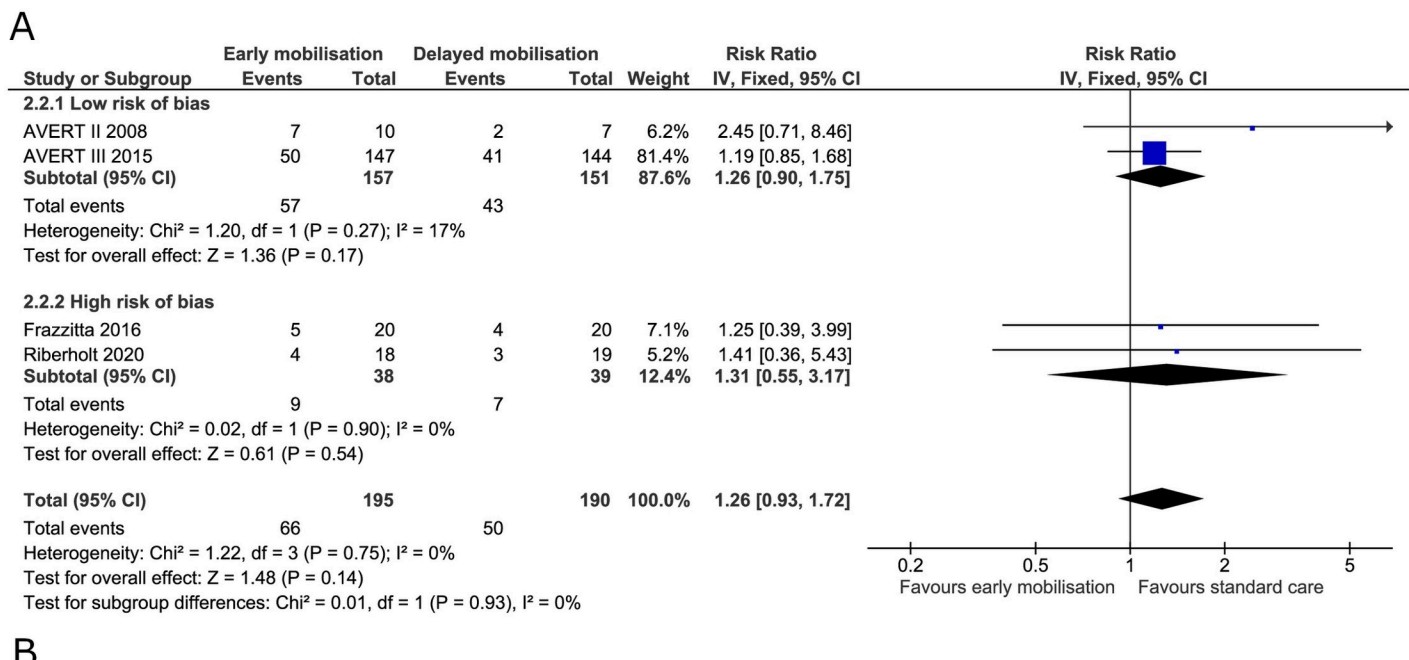

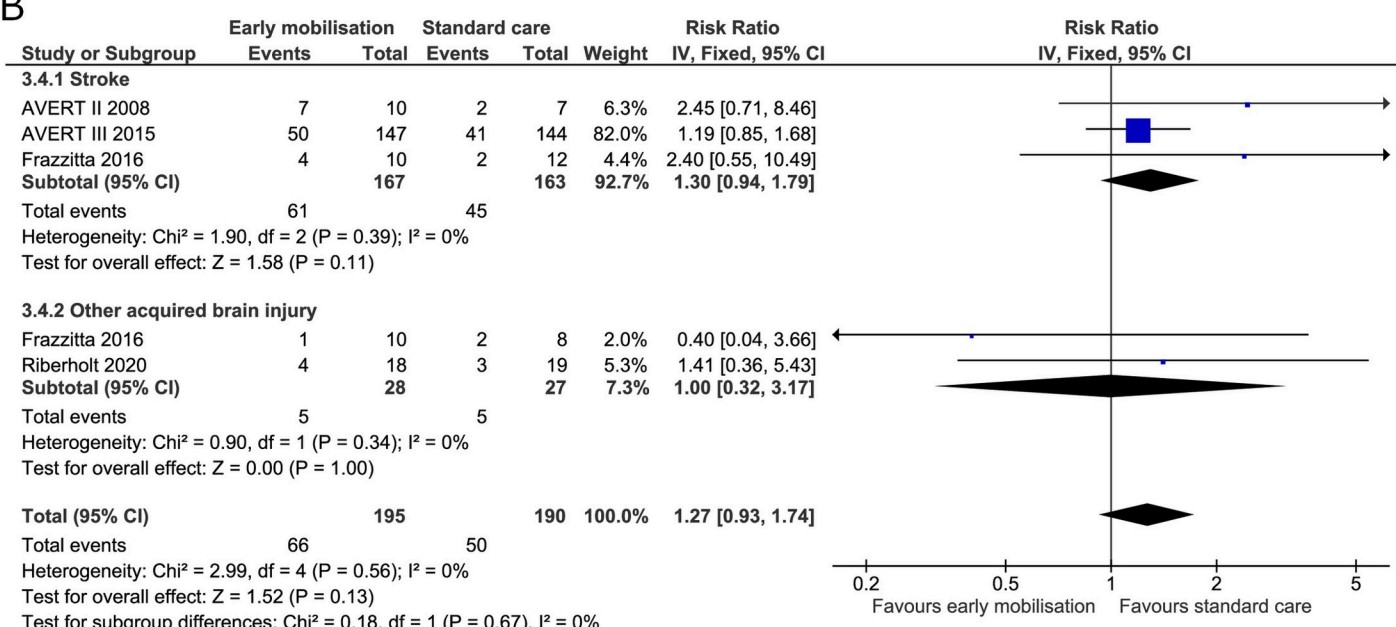

**Fig 7. Comparison of early mobilisation versus standard care–mortality at maximal follow-up.** Fig 7A and 7B. Forest-plots showing the results from the fixed-effect meta-analysis of the outcome mortality at maximal follow-up with subgroup divided according to risk of bias (A) or diagnosis (B).

## Exploratory outcomes

Three trials reported data on other adverse events besides death [5,19,36]. The RR of patients experiencing serious adverse events and adverse events not considered serious in the early mobilisation group versus the standard care group can be found in S2 Table in S1 File. For death, please see analysis in Figs 6 and 7. Because there were no events in the standard care group, we calculated the RR with a 0.5 event correction. The results from the fixed-effect model yielded a RR of 4.43 for acute myocardial infarction (95% CI 0.4 to 51.5) and a RR of 13.0 for confusion (95% CI 0.2 to 723.5).

A

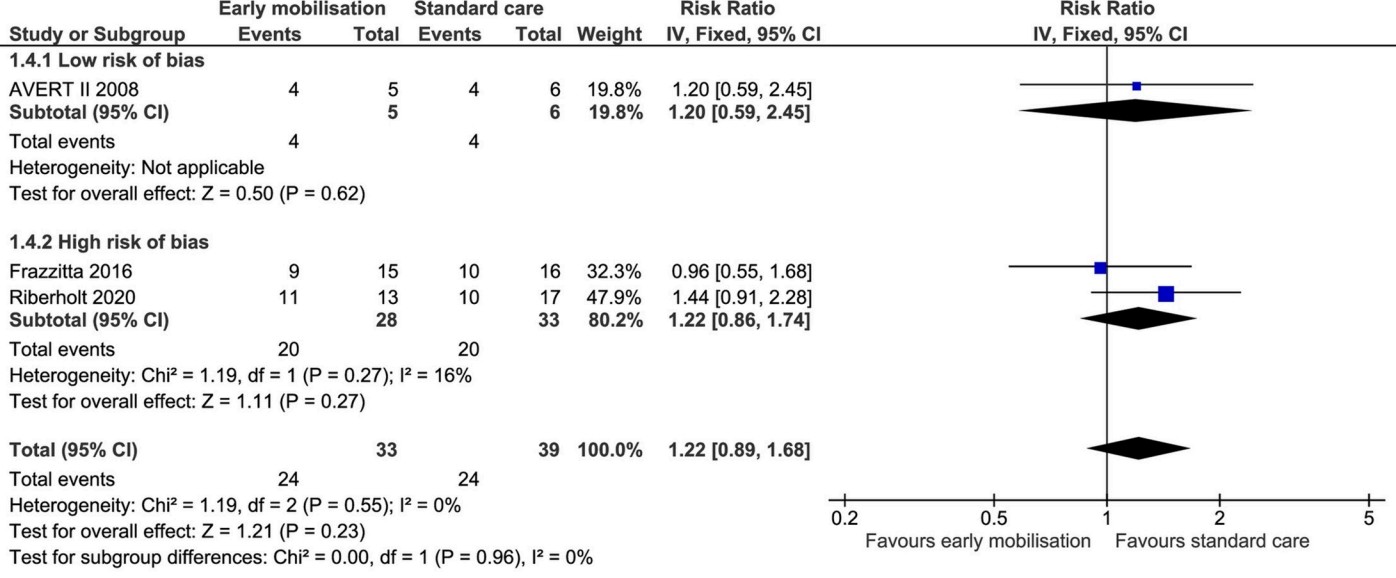

B

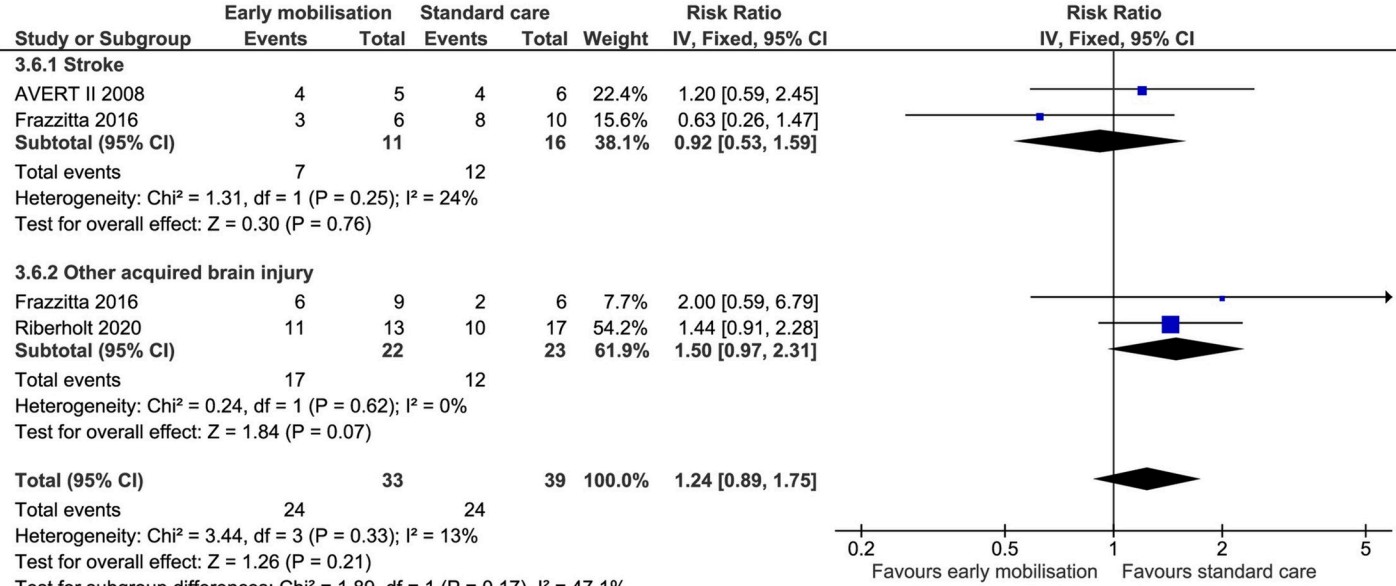

**Fig 8. Comparison of early mobilisation versus standard care–Poor functional outcome among survivors at the end of the intervention.** Fig 8A and 8B. Forest-plots showing the results from the fixed-effect meta-analysis of the outcome poor function at the end of intervention with subgroup divided according to risk of bias (A) or diagnosis (B).

**Adverse events in observational studies.** We were able to retrieve data on serious adverse events and adverse events not considered serious from the observational case-control study by Karic et al. 2016 [71]. This study included 25 patients with severe subarachnoid haematoma (Hunt and Hess grade IV or V) [72]. Serious adverse events occurred in 11/14 patients in the early mobilisation group compared to 10/11 patients in the standard care group. Adverse events not considered serious were reported in 13/14 of patients in the early mobilisation group compared to 10/11 in the standard care group. These events were not described further.

## Summary of findings

Please see Table 3.

## Discussion

### Summary of main results

We identified four trials assessing early mobilisation compared with standard care. In general, the sample size was far from large enough to make firm conclusions on the benefits or harms

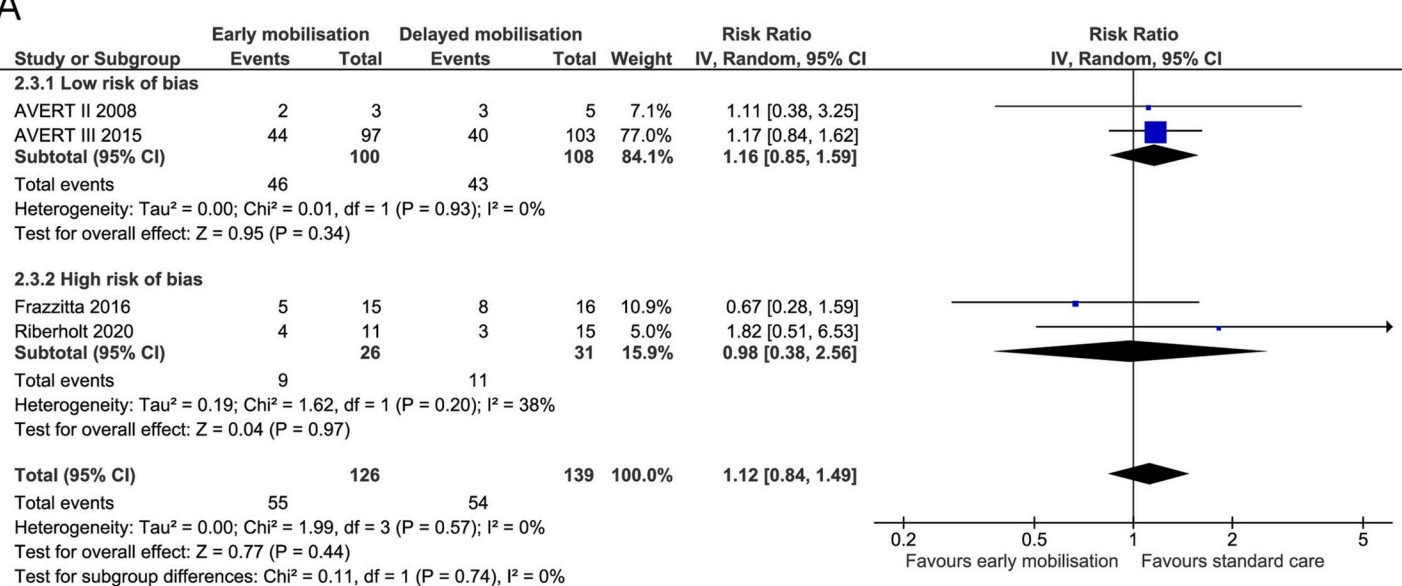

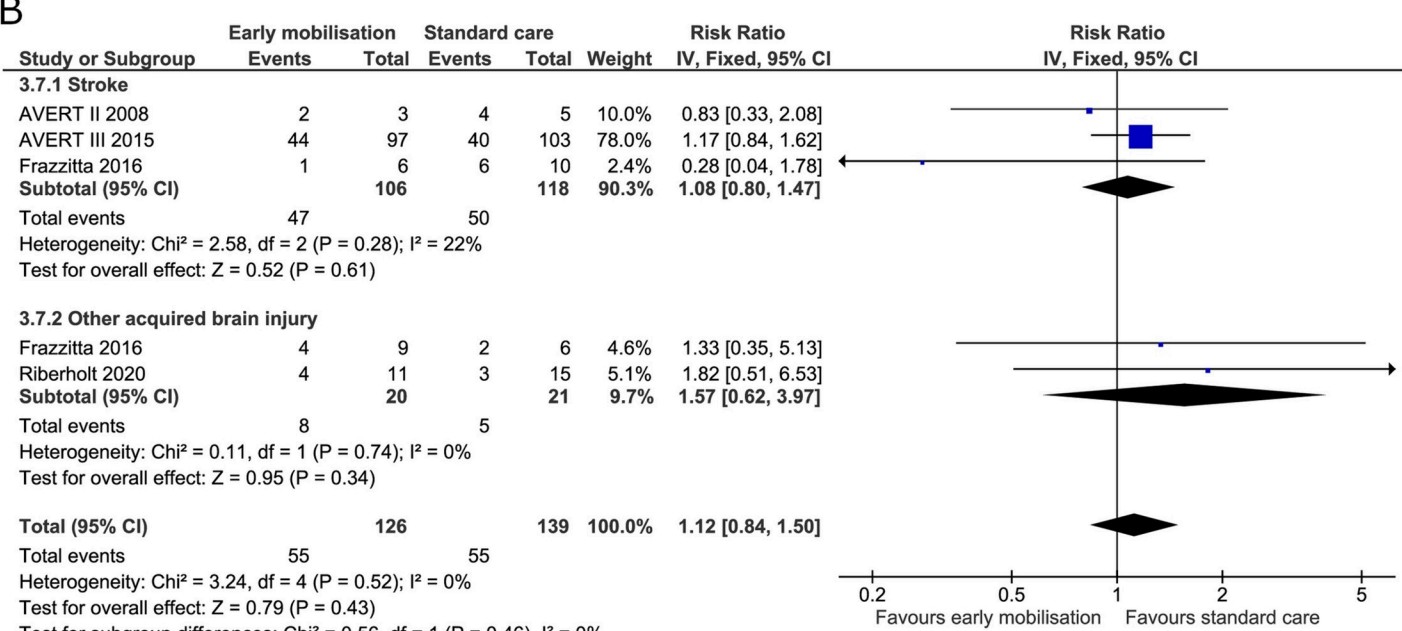

**Fig 9. Comparison of early mobilisation versus standard care–Poor functional outcome among survivors at maximal follow-up.** Figs 9A and 8B. Forest-plot showing the results from the fixed-effect meta-analysis of the outcome poor functional outcome at the longest follow-up with subgroup divided according to risk of bias (A) or diagnosis (B).

A

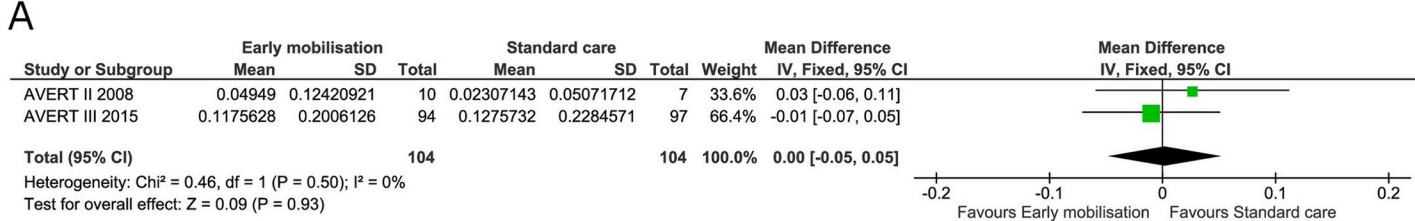

**Fig 10. Comparison of early mobilisation versus standard care–Quality of life at maximal follow-up.** Fig 10 A. Forest-plot showing the results from the fixed-effect meta-analysis of the outcome quality of life at maximal follow-up with subgroup divided according to risk of bias.

of early mobilisation. Thus, no difference between early mobilisation or standard care was found on death or poor functional outcome either at the end of intervention or at maximal follow-up. In patients with a severe stroke, there seems to be enough evidence to conclude that early mobilisation does not change the quality of life. However, given the moderate certainty of the evidence and the fact that this conclusion was based on a subgroup analysis, this result should be confirmed by future trials. Serious adverse events were almost evenly divided between intervention groups but given that only 18% to 24% of the required information size was reached more trials are needed. However, we noted increased risk of acute myocardial infarction and confusion in the experimental group. It should be emphasised that the small number of trials and patients in this review only provides limited evidence.

Early intervention was started far sooner in the two trials including patients with severe stroke [5,19] than in the two trials including patients with severe traumatic brain injury [36,70]. The type of mobilisation used also differed, as the two latter trials used a tilt table whereas the former used manual mobilisation to the edge of bed/chair or standing position if possible. However, these differences did not seem to affect the estimated intervention effects as we found no heterogeneity and no difference in subgroup analyses.

The subgroup analysis between patients with stroke and non-stroke acquired brain injury showed no difference in the estimates of the primary outcomes.

The other outcomes investigated in this trial did not reach the boundaries for futility but required larger information sizes (between 393 and 4,342). Thus, this review strongly indicates a need for more research on patients with severe acquired brain injury to answer questions on the effects of early mobilisation.

## Overall completeness and applicability of evidence

Our search was comprehensive and employed an inclusive approach. Besides searching medical databases, we also searched clinical trial registries and grey literature (Google Scholar etc.). We contacted 17 primary investigators of other trials with experience in the field of early mobilisation and brain injury. The six non-responders could potentially have trials that fitted our inclusion criteria.

There is a clear growing interest in early mobilisation in other patients than the stroke population, although the latter constituted the majority in this review. Besides 'wrong study design' (observational or quasi-randomised), 'not doing early mobilisation' but rather other early interventions and 'wrong patient population' were the main reasons for excluding studies. Thus, one very large trial [16] using elevation of the head of the bed to 30 degrees did not fulfil our inclusion criteria.

In general, information about the effect of mobilising in patients with severe acquired brain injury is limited. Even when we combine patients with different types of brain injury (which

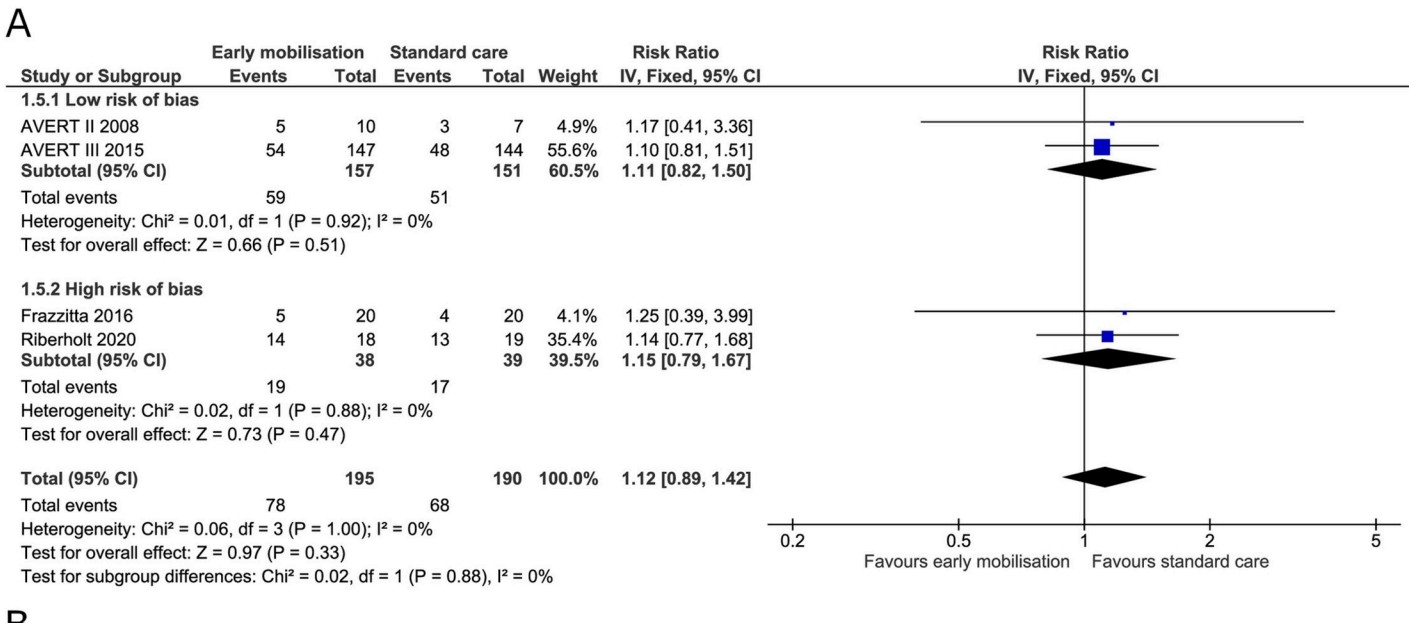

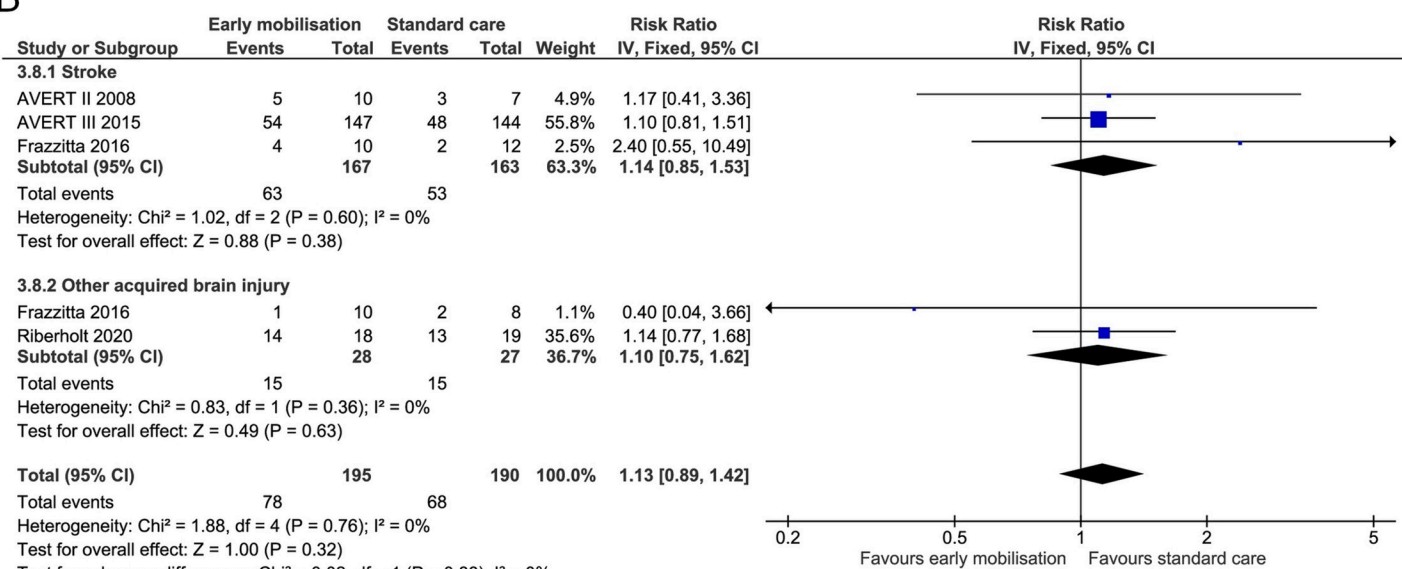

**Fig 11. Comparison of early mobilisation versus standard care–Serious adverse events at the end of the intervention.** Fig 11 A and B. Forest-plots showing the results from the fixed-effect meta-analysis of the outcome serious adverse events at the end of intervention with subgroup divided according to risk of bias (A) or diagnosis (B).

could be considered going too far), the amount of data is still sparse within every single pathology.

## Quality of the evidence

All four trials used random sequence generation and one trial showed risk of attrition bias [36]. Two trials attempted to blind patients and staff [5,19] by not revealing details of the treatment protocols and providing the treatment behind curtains so the risk of bias was reduced. The staff and the patients in two trials were not blinded to the protocol or treatment allocation [36,70]. One could argue that the patients had low enough Glasgow Coma Score to be

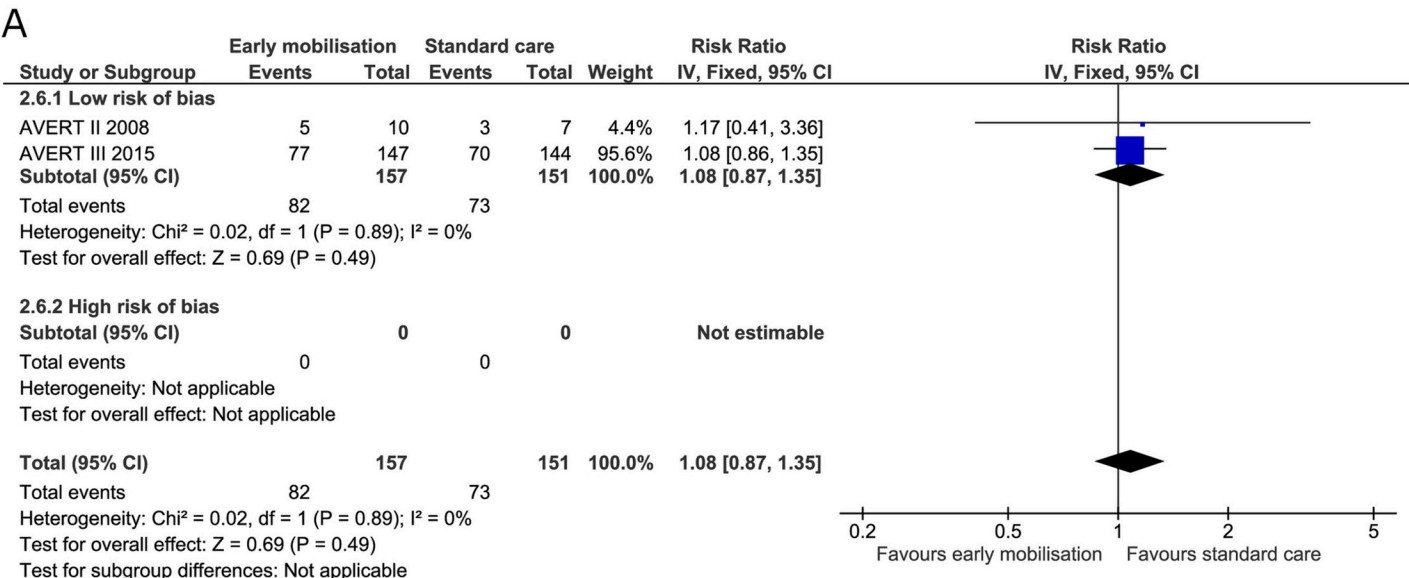

**Fig 12. Comparison of early mobilisation versus standard care–Serious adverse events at maximal follow-up.** Fig 12 A. Forest-plot showing the results from the fixed-effect meta-analysis of the outcome serious adverse events at the longest follow-up with subgroup divided according to low risk of bias or risk of bias.

considered indirectly blinded. Blinded outcome assessors were used in different degree in all four trials. The AVERT trials [5,19] blinded assessors for all outcomes, the trial by Frazzitta et al. blinded for all outcomes except for all adverse events [70] while the last trial [36] only blinded for two outcomes (Coma Recovery Scale-Revised and adverse events). Risk of selective outcome reporting was not possible to assess in one trial [70] since the trial was retrospectively reported to clinicaltrials.gov. There were some differences in baseline characteristics in the two small trials [36,70] in that those in the early mobilisation group were older in one study and younger in the other.

The homogeneity of the interventions applied in the included studies could be questioned. There was a large variety of the definition of early mobilisation ranging from hours to weeks after injury. Nevertheless, some common ground can be found. Thus, all trials attempted to administer mobilisation at an earlier time than was standard for the respective patient groups. This should be considered when interpreting the findings in the context of clinical practice.

## Potential biases in the review process (strengths and limitations)

The strength of this review is that we published the protocol on PROSPERO (CRD42018088790) before searching. We conducted thorough searches on relevant databases, trial registries, searched for unpublished data and a wide search strategy was applied for the intervention 'early mobilisation' as the term is poorly defined. We also used a rigorous selection procedure by only including randomised clinical trials where at least 10 patients matched our inclusion criteria. All included studies were assessed using Cochrane's Risk of Bias Tool and the quality was assessed using the GRADE methodology. Furthermore, we utilised the TSA to control for random errors.

Chinese- and Russian-language trials were translated into English before the full text was assessed for relevance. Some discrepancies could occur during the translation process, which may have affected the decision of exclusion. Also, the interventions and patient populations were somewhat loosely described in Chinese-language trials, probably due to cultural differences.

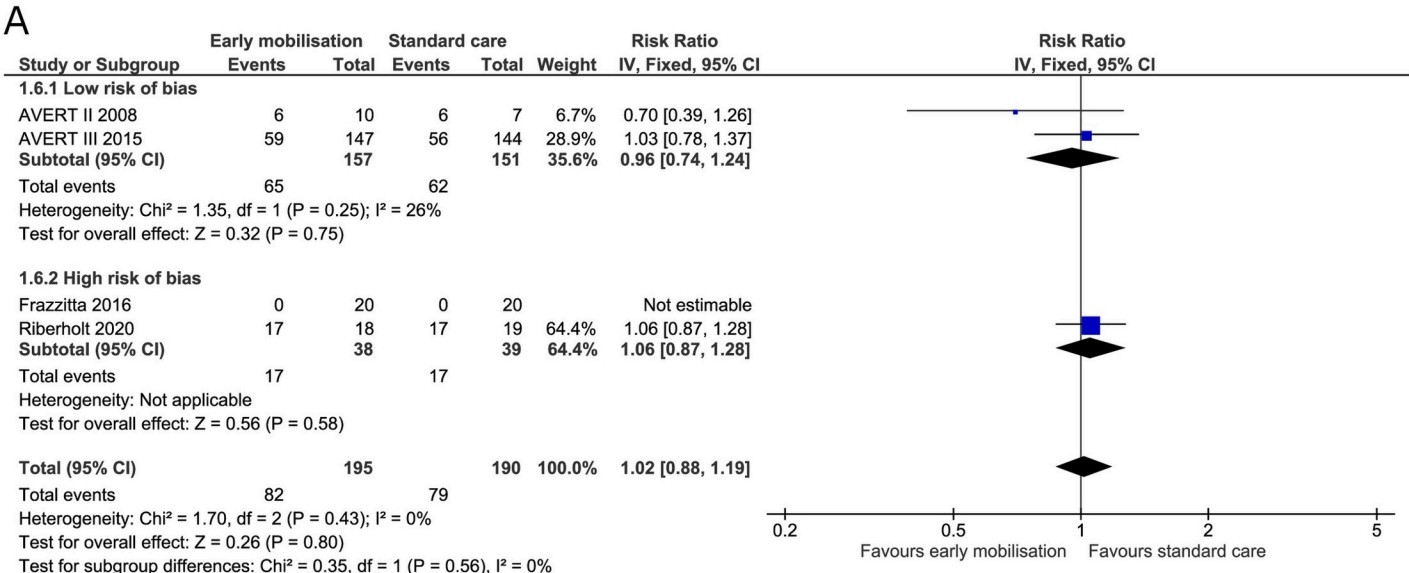

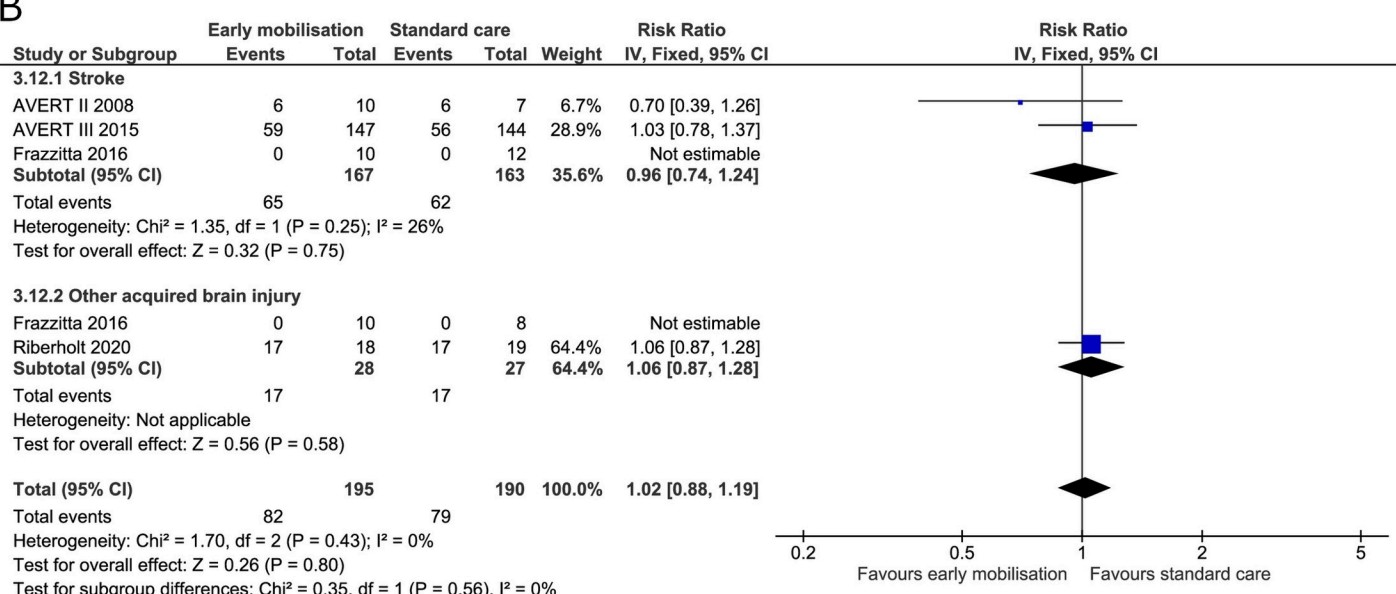

**Fig 13. Comparison of early mobilisation versus standard care–Coma Recovery Scale-Revised at the end of the intervention.** Fig 13A and 13B. Forest-plot showing the results from the random-effects meta-analysis of the outcome Coma Recovery Scale-Revised at the end of intervention with subgroup divided according to risk of bias (A) or diagnosis (B).

Due to our selection of designs in this review our chances of discovering harms were limited, especially long-term and exceedingly rare harms resulting from the interventions. Therefore, we decided to present the characteristics of serious adverse events and adverse events not considered serious and to include observational studies for the investigation of harms. Because we applied a "randomised clinical trial"-filter in our search, only one relevant study was identified.

The results of this review are highly driven by the data from the AVERT III trial and, therefore, mostly reflects the effect of the intervention on patients with severe stroke. The younger population of patients with traumatic brain injury could react differently to this intervention,

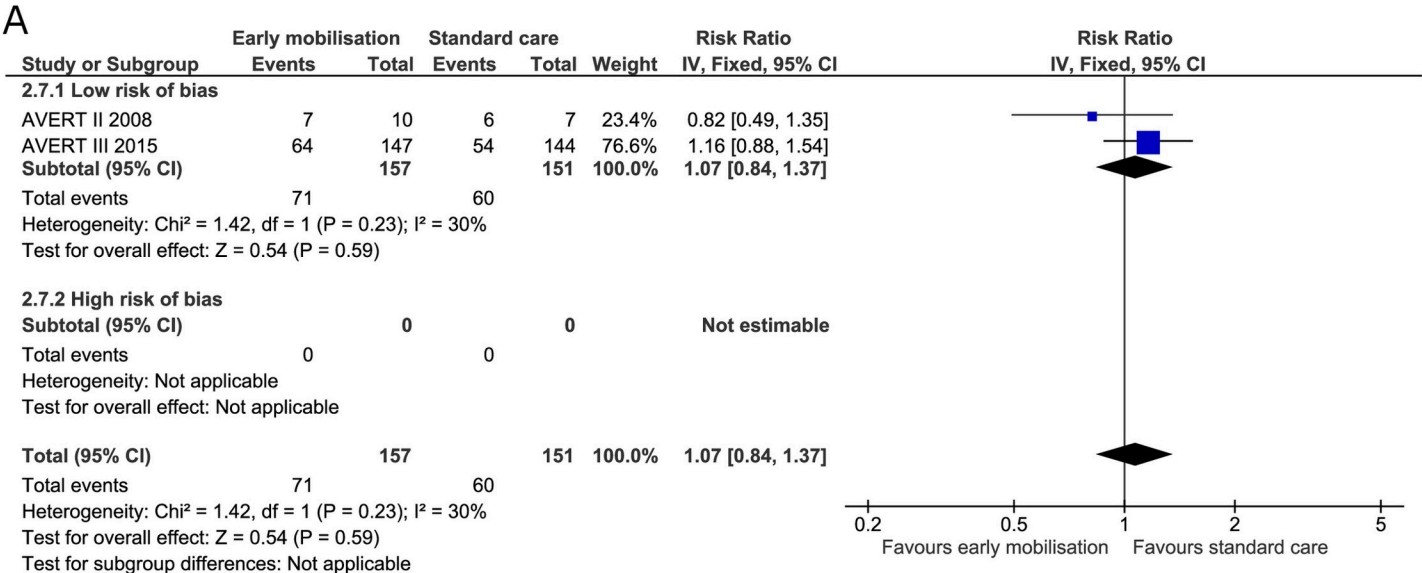

**Fig 14. Comparison of early mobilisation versus standard care–Coma Recovery Scale-Revised at maximal follow-up.** Fig 14A and 14B. Forest-plot showing the results from the random-effects meta-analysis of the outcome Coma Recovery Scale-Revised at the longest follow-up with subgroup divided according to risk of bias (A) or diagnosis (B).

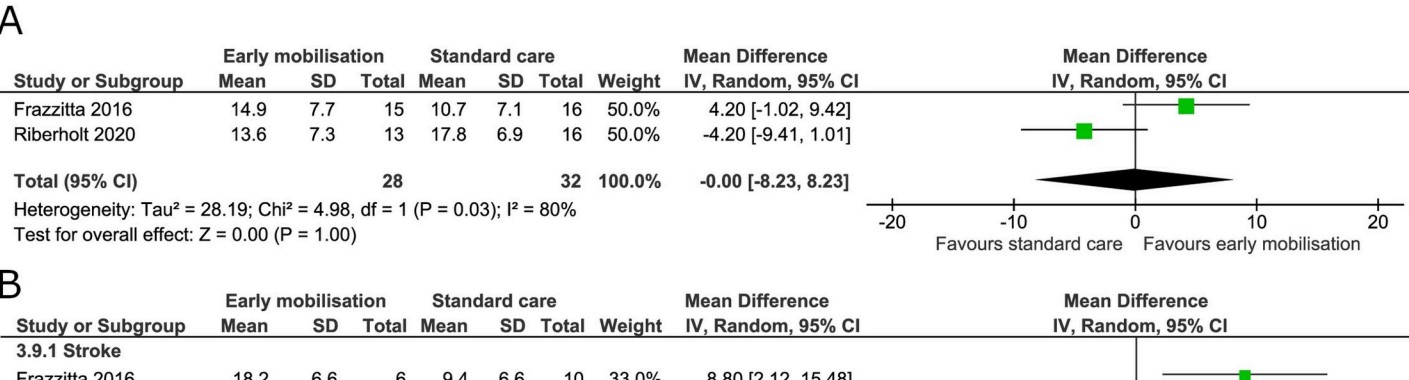

**Fig 15. Comparison of early mobilisation versus standard care–Adverse events not considered serious at the end of the intervention.** Fig 15A and 15B. Forest-plot showing the results from the fixed-effect meta-analysis of the outcome adverse events not considered serious at the end of intervention with subgroup divided according to risk of bias (A) or diagnosis (B).

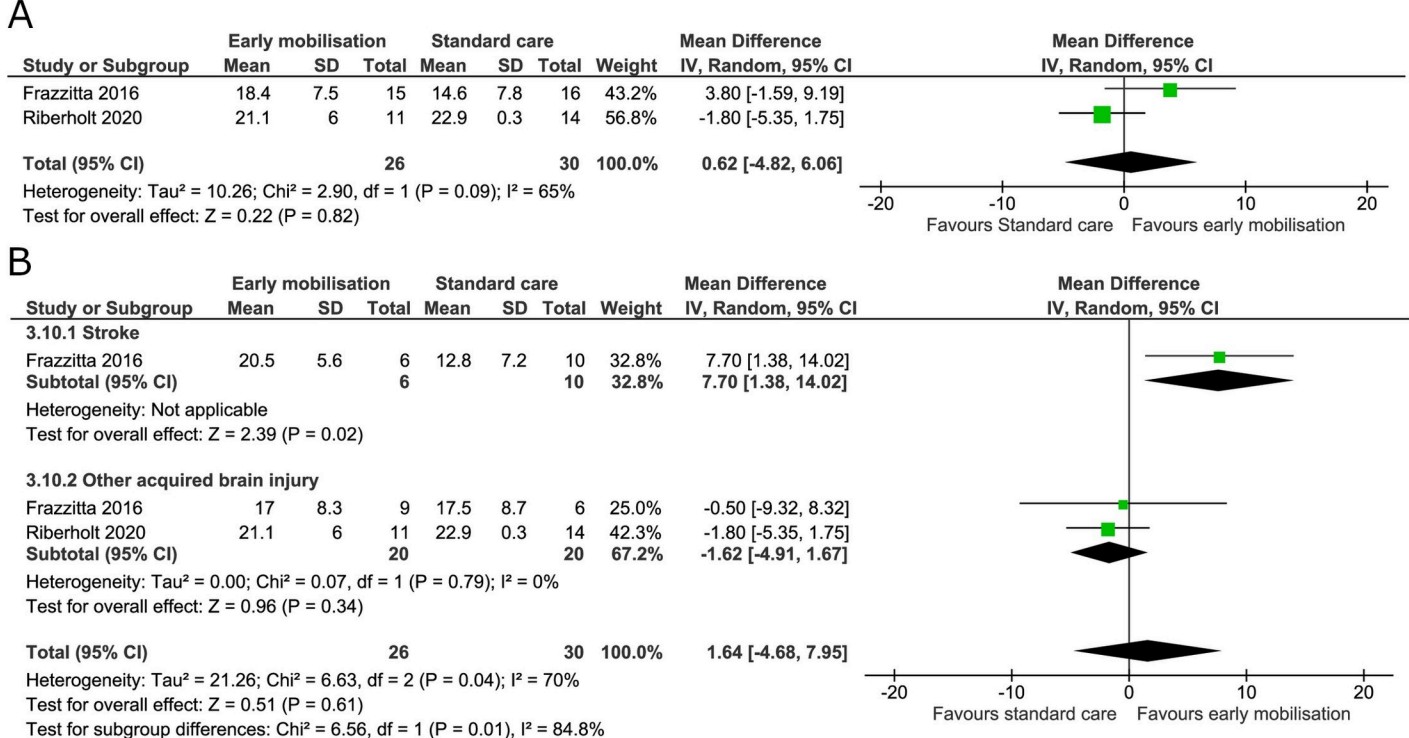

**Fig 16. Comparison of early mobilisation versus standard care–Adverse events not considered serious at maximal follow-up.** Fig 16A. Forest-plot showing the results from the fixed-effect meta-analysis of the outcome adverse events not considered serious at the maximal follow-up with subgroup divided according to risk of bias.

**Table 3. Summary of findings for early mobilisation versus delayed mobilisation.**

**Early mobilisation compared with standard care (delayed mobilisation) for patients with severe acquired brain injury**

**Patient or population**: patients with severe acquired brain injury including traumatic brain injury, stroke, anoxic brain injury

**Setting**: stroke unit or critical care unit

**Intervention**: early mobilisation

**Comparison**: standard care (delayed mobilisation)

| Outcomes (Importance) | Anticipated absolute effects* (95% CI) | | Relative effect (95% CI) | № of participants (studies) | Certainty of the evidence (GRADE) | Comments |
|---|---|---|---|---|---|---|
| | Risk with control | Risk with early mobilisation | | | | |
| Death or poor functional outcome at the end of the intervention (Critical) | 565 per 1.000 | **673 per 1.000** (526 to 865) | **RR 1.19** (0.93 to 1.53) | 91 (3 RCTs) | ⊕○○○VERY LOW [a, b] | Large difference in the populations between trials, although we did not observe subgroup differences regrading intervention effects. Small number of patients in trials. Large beneficial effects to harmful effects are possible according to the TSA-adjusted CI. |
| Death or poor functional outcome at the maximal follow-up (Critical) | 603 per 1.000 | **621 per 1.000** (537 to 730) | **RR 1.03** (0.89 to 1.21) | 381 (4 RCTs) | ⊕○○○ VERY LOW [a, b] | Large difference in the populations between trials, although we did not observe subgroup differences regrading intervention effects. Beneficial effects to harmful effects are possible according to the TSA-adjusted CI. |

*(Continued)*

**Table 3.** (Continued)

**Early mobilisation compared with standard care (delayed mobilisation) for patients with severe acquired brain injury**

| | | | | | | |
|---|---|---|---|---|---|---|
| Quality of Life at the maximal follow-up assessed with: AQoL (4D) (Critical) | The mean quality of Life was **0** AQoL points | MD **0 AQoL points** (0.05 lower to 0.05 higher) | - | 208 (2 RCTs) | ⊕⊕⊕○ MODERATE[x] | Patients included in this analysis are from trials only including patients with severe stroke. |
| Patients with at least one serious adverse event at the end of the intervention (Critical) | 358 per 1.000 | **394 per 1.000** (308 to 497) | **RR 1.10** (0.86 to 1.39) | 385 (4 RCTs) | ⊕○○○ VERY LOW [q, b] | Large difference in the populations between trials, although we did not observe subgroup differences regrading intervention effects. Beneficial effects to harmful effects are possible according to the TSA-adjusted CI. |
| Patients with at least one serious adverse event at the maximal follow-up (Critical) | 483 per 1.000 | **522 per 1.000** (421 to 653) | **RR 1.08** (0.87 to 1.35) | 308 (2 RCTs) | ⊕○○○ VERY LOW [q, b] | Patients included in this analysis are from trials only including patients with severe stroke. Beneficial effects to harmful effects are possible according to the TSA-adjusted CI. |
| Coma Recovery Scale-Revised (CRS-R) at end of the intervention (Critical) | The mean coma Recovery Scale-Revised (CRS-R) at end of the intervention was **-0.00** CRS-R points | MD **0 CRS-R points** (8.23 lower to 8.23 higher) | - | 60 (2 RCTs) | ⊕○○○ VERY LOW [b, e, f] | The two trials have high risk of bias. There is inconsistency between the two trials estimates. Large beneficial effects to harmful effects are possible according to the TSA-adjusted CI. |
| Coma Recovery Scale-Revised (CRS-R) at the maximal follow-up (Critical) | The mean coma Recovery Scale-Revised (CRS-R) at the maximal follow-up (Random) was **0.62** CRS-R points | MD **0.62 CRS-R points higher** (4.82 lower to 6.06 higher) | - | 56 (2 RCTs) | ⊕○○○ VERY LOW [b, e, f] | The two trials have high risk of bias. There is inconsistency between the two trials estimates. Extreme beneficial effects to harmful effects are possible according to the TSA-adjusted CI. |

*The risk in the intervention group** (and its 95% confidence interval) is based on the assumed risk in the comparison group and the **relative effect** of the intervention (and its 95% CI). **CI:** Confidence interval; **RR:** Risk ratio; **MD:** Mean difference

GRADE Working Group grades of evidence

High certainty: We are very confident that the true effect lies close to that of the estimate of the effect

Moderate certainty: We are moderately confident in the effect estimate: The true effect is likely to be close to the estimate of the effect, but there is a possibility that it is substantially different

Low certainty: Our confidence in the effect estimate is limited: The true effect may be substantially different from the estimate of the effect

Very low certainty: We have very little confidence in the effect estimate: The true effect is likely to be substantially different from the estimate of effect

a. Downgraded for indirectness by one level due to differences in outcome measures for measuring poor functional outcome; b. Downgraded for imprecision by two levels as the accrued information size was below 50% of the diversity-adjusted required information size; c. downgraded one level due to indirectness as the trials only consist of patients with stroke. d. Downgraded for indirectness by one level due to differences in intervention. e. Downgraded one level for high risk of bias as both included trials was at high risk of bias; f. Downgraded one level for inconsistency as considerable to substantial heterogeneity was found.

although they are a very heterogeneous group. More trials are needed to draw firm conclusions, for younger patients and traumatic brain injury.

The four included trials used different outcome scales for our primary outcome exploring physical function. Therefore, we dichotomised these outcomes as a poor or good outcome and this incurs a risk of losing information. The analysis of the results can then be greatly affected by the distribution of the outcome and the specific cut-off between 'poor' or 'good'. Furthermore, the statistical power of the analysis is lower when dichotomising a continuous scale [73,74]. We dichotomised the mRS with 5 and 6 as a poor outcome and 1 to 4 as a good outcome. This could be considered somewhat uncommon. But given that we are including patients with severe brain injury it could also be considered successful to move patients to a better outcome than 5 or 6. Alternatively, we could have used the standardised mean difference

to analyse different outcome measures used to assess physical function, but we believe this method can be hard to interpret for clinical relevance.

We were only able to complete the subgroup analysis exploring the low risk of bias versus the high risk of bias and one of the planned clinical subgroup analyses (stroke versus other brain injuries). The included trials did not differ enough in duration, intensity, frequency, timing or type of mobilisation to make these analyses relevant.

The small sample sizes in this review were exposed when compared to the TSA required information sizes for the different selected outcomes. Future trials are needed to gain sufficient knowledge about the benefits and harms of the treatment, but will most likely increase the heterogeneity and the required information size would increase accordingly [75].

Because of the low number of included trials in this review, it was not rational to make a funnel plot for analysing publication bias. The published randomised controlled trials were few, and future studies should emphasise blinding participants, personnel, and outcome assessments to avoid downgrading the certainty of evidence.

## Agreements and disagreements with other studies or reviews

No other review has undertaken the challenge of investigating the effect of early mobilisation in patients with severe acquired brain injury. A Cochrane review on patients with stroke and early mobilisation found no benefits of early mobilisation on the number of people who survived or made a good recovery [76], but patients with severe stroke are underreported in these trials, even though 14% of the patients in the single largest study [5] had a severe stroke. Interestingly, these results are very similar to the results in the present review on the outcome of death or poor function at three months [76]. It could, therefore, be hypothesised that patients with severe stroke (NIHSS>16) experience the same harms from an early mobilisation as do other patients with stroke. Likewise, a recently published analysis from the AVERT III trial showed that quality of life is not improved from early mobilisation, which aligns with our results [77].

Another recently published review found benefits on functional recovery of early rehabilitation interventions in patients with traumatic brain injury starting at the trauma centre and more intensive neurorehabilitation afterwards [78]. The included studies were small (largest n = 86) and included a quasi-randomised trial, which could lead to a risk of type I or II errors. Trials investigating complicated rehabilitation programs such as 'systematic reality orientation program' or 'multisensory stimulation' can be difficult to replicate in a rigid clinical trial and therefore also difficult to transform into a clinical setting.

This review differs from other reviews because patients with severe acquired brain injury were specifically selected for analysis. In the search for more homogenous patient populations, the patients with severe brain injury have often been excluded from other randomised clinical trials of early mobilisation in critically ill patients.

## Authors' conclusions

### Implications for practice

We found no evidence of a difference between the early mobilisation of patients with severe acquired brain injury compared with standard care in important outcomes such as death and poor functional outcome, or serious adverse events. Our analyses also do not indicate a major impact on the quality of life as measured with AQoL(4D), although smaller effects and effects on other measures of life quality cannot be excluded. Our systematic review strongly highlights the insufficient evidence in patients with severe brain injury, and no firm conclusions on the potential benefit or harm from early mobilisation can be drawn from these data.

## Implications for research

More research is needed within the area of early mobilisation for severe brain injury, especially in the subgroups of participants, namely patients with traumatic brain injury, stroke, and hypoxic brain injury. Outcomes such as the effect on death alone or functional outcome alone and quality of life in other patients with a brain injury than stroke, as well as harms (serious adverse events) should be further investigated. Future trials should closely monitor patients for potential adverse events like AMI and confusion.

## Supporting information

**S1 File.**
(PDF)

## Acknowledgments

We would like to thank Information specialist Sarah Louise Klingenberg at The Cochrane Hepato-Biliary Group, Copenhagen for help in developing the search strategy.

## Author Contributions

**Conceptualization:** Christian Gunge Riberholt, Jane Lindschou, Christian Gluud, Jesper Mehlsen, Kirsten Møller.

**Data curation:** Christian Gunge Riberholt, Vibeke Wagner, Jane Lindschou.

**Formal analysis:** Christian Gunge Riberholt.

**Funding acquisition:** Christian Gunge Riberholt.

**Investigation:** Christian Gunge Riberholt, Vibeke Wagner, Jane Lindschou.

**Methodology:** Christian Gunge Riberholt, Jane Lindschou, Christian Gluud, Jesper Mehlsen, Kirsten Møller.

**Project administration:** Christian Gunge Riberholt.

**Resources:** Vibeke Wagner.

**Supervision:** Christian Gluud.

**Visualization:** Christian Gunge Riberholt.

**Writing – original draft:** Christian Gunge Riberholt.

**Writing – review & editing:** Christian Gunge Riberholt, Vibeke Wagner, Jane Lindschou, Christian Gluud, Jesper Mehlsen, Kirsten Møller.

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
