## [Decision Letter · Decision Letter 0]

12 Jun 2020

PONE-D-20-13423

Early head-up mobilisation versus standard care for patients with severe acquired brain injury: a systematic review with meta-analysis and Trial Sequential Analysis

PLOS ONE

Dear Dr. Riberholt,

Thank you for submitting your manuscript to PLOS ONE. After careful consideration, we feel that it has merit but does not fully meet PLOS ONE’s publication criteria as it currently stands. Therefore, we invite you to submit a revised version of the manuscript that addresses the points raised during the review process.

We look forward to receiving your revised manuscript.

Kind regards,

Ivan D. Florez

Academic Editor

PLOS ONE

Journal Requirements:

Reviewers' comments:

Reviewer's Responses to Questions

**Comments to the Author**

1. Is the manuscript technically sound, and do the data support the conclusions?

Reviewer #1: Yes

Reviewer #2: Yes

2. Has the statistical analysis been performed appropriately and rigorously? 

Reviewer #1: Yes

Reviewer #2: Yes

3. Have the authors made all data underlying the findings in their manuscript fully available?

Reviewer #1: Yes

Reviewer #2: Yes

4. Is the manuscript presented in an intelligible fashion and written in standard English?

Reviewer #1: Yes

Reviewer #2: Yes

5. Review Comments to the Author

Reviewer #1: Please note: I am commenting only on review and statistical methods; I cannot comment on the clinical value of the paper.

In general, I found this to be a thorough and well-conducted systematic review. I did not find any area where anything was missing, or any errors in analysis or conclusions.

I note that the review includes only 4 trials and 385 patients. It should be clearly acknowledged that this is a rather limited evidence base. I am concerned that it cannot really support the large number of analyses and outcomes considered, particularly in the level of detail reported, and perhaps some simplification would be appropriate; for example, to move exploratory outcomes to an appendix.

The paper is rather repetitive in places, particularly (but not only) the many “The RR of”s in the exploratory outcomes section. In general, there is no need to repeat in text what is reported in Tables or figures, and simply presenting tables of results (and forest plots), with a short text commentary, would make the paper an easier read.

While I appreciate the use of TSA, I do not think it is really adding much to this paper: the evidence base is clearly too small, and the effect estimates too close to the null, for it to produce any useful conclusions. I note, for example, that in most TSA plots there is too little data even to establish the alpha spending boundaries. Perhaps the TSA analyses should be placed in an appendix, to make the paper shorter and easier to follow for readers not familiar with TSA.

Reviewer #2: General comments:

Based on the PLOS-ONE guidelines for authors, tables should not be within the text. We suggest the authors to arrange the document as mentioned in PLOS-ONE guidelines.

The authors included observational data as part of the results, however, there is no information about how the risk of bias and certainty of the evidence was assessed from these sources.

Abstract;

Abstract may be improved by following the PRISMA recommendations for abstracts.

Methods:

Sensitivity analysis can be described as separated section.

GRADE description may be improved, for example, to report the outcomes importance (critical, important, non important)

Non-randomized data

How the observational data was assessed?

Results:

PLOS-ONE does not limit the number of characters in their manuscripts, however, the document feels very saturated of information that may be reported in a more structure way. If authors are using the GRADE approach, to describe the results by the importance of the outcomes might be a feasible idea.

Exploratory outcomes section does not have any kind of arrangement, sorting the evidence by type of outcome or other approach might help to improve this section.

GRADE: More information needs to be provided in the GRADE explanations footnotes (Summary of Finding table). For example, We decided to downgrade by one level due to imprecision, because the effect estimate was not consistent with benefit or harm, or other kind of judgment that can help the reader to understand the reasoning behind the decision.

The quality of the figures (Forest plots and sequential analysis) was very poor, this should be improved - for some of the figures was impossible to read the results.

6. PLOS authors have the option to publish the peer review history of their article (what does this mean?). If published, this will include your full peer review and any attached files.

Reviewer #1: No

Reviewer #2: Yes: Colunga-Lozano Luis Enrique

---

## [Author Response · Author response to Decision Letter 0]

28 Jun 2020

Please see uploaded file "2020_06_28_Point_by_point response.doc"

---

## [Decision Letter · Decision Letter 1]

22 Jul 2020

Early head-up mobilisation versus standard care for patients with severe acquired brain injury: a systematic review with meta-analysis and Trial Sequential Analysis

PONE-D-20-13423R1

Dear Dr. Riberholt,

We’re pleased to inform you that your manuscript has been judged scientifically suitable for publication and will be formally accepted for publication once it meets all outstanding technical requirements.

Kind regards,

Ivan D. Florez

Academic Editor

PLOS ONE

Additional Editor Comments (optional):

You have addressed all reviewers' comments and concerns.

Reviewers' comments:

Reviewer's Responses to Questions

**Comments to the Author**

1. If the authors have adequately addressed your comments raised in a previous round of review and you feel that this manuscript is now acceptable for publication, you may indicate that here to bypass the “Comments to the Author” section, enter your conflict of interest statement in the “Confidential to Editor” section, and submit your "Accept" recommendation.

Reviewer #2: All comments have been addressed

2. Is the manuscript technically sound, and do the data support the conclusions?

Reviewer #2: Yes

3. Has the statistical analysis been performed appropriately and rigorously? 

Reviewer #2: Yes

4. Have the authors made all data underlying the findings in their manuscript fully available?

Reviewer #2: Yes

5. Is the manuscript presented in an intelligible fashion and written in standard English?

Reviewer #2: Yes

6. Review Comments to the Author

Reviewer #2: (No Response)

7. PLOS authors have the option to publish the peer review history of their article (what does this mean?). If published, this will include your full peer review and any attached files.

Reviewer #2: **Yes: **Luis Enrique Colunga Lozano

---

## [Editor Report · Acceptance letter]

29 Jul 2020

PONE-D-20-13423R1 

Early head-up mobilisation versus standard care for patients with severe acquired brain injury: a systematic review with meta-analysis and Trial Sequential Analysis 

Dear Dr. Riberholt:

I'm pleased to inform you that your manuscript has been deemed suitable for publication in PLOS ONE. Congratulations! Your manuscript is now with our production department. 

Kind regards, 

on behalf of

Dr. Ivan D. Florez 

Academic Editor

PLOS ONE